# A Simple Remedy for Dataset Bias via Self-Influence: A Mislabeled Sample Perspective

**Yeonsung Jung**[*1], **Jaeyun Song**[*1], **June Yong Yang**[1]
**Jin-Hwa Kim**[2,3], **Sung-Yub Kim**[1], **Eunho Yang**[1,4]
[1]Korea Advanced Institute of Science and Technology, [2]NAVER AI Lab
[3]Seoul National University, [4]AITRICS
{ys.jung, mercery}@kaist.ac.kr[1]

## Abstract

Learning generalized models from biased data is an important undertaking toward fairness in deep learning. To address this issue, recent studies attempt to identify and leverage bias-conflicting samples free from spurious correlations without prior knowledge of bias or an unbiased set. However, spurious correlation remains an ongoing challenge, primarily due to the difficulty in precisely detecting these samples. In this paper, inspired by the similarities between mislabeled samples and bias-conflicting samples, we approach this challenge from a novel perspective of mislabeled sample detection. Specifically, we delve into Influence Function, one of the standard methods for mislabeled sample detection, for identifying bias-conflicting samples and propose a simple yet effective remedy for biased models by leveraging them. Through comprehensive analysis and experiments on diverse datasets, we demonstrate that our new perspective can boost the precision of detection and rectify biased models effectively. Furthermore, our approach is complementary to existing methods, showing performance improvement even when applied to models that have already undergone recent debiasing techniques.

## 1 Introduction

Deep neural networks have demonstrated remarkable performance in various fields of machine learning tasks comparable to or superior to humans on well-curated benchmark datasets [7, 4, 60, 14]. Nevertheless, the efficacy of these models trained on unfiltered, real-world data remains an open question. In this scenario, a significant concern arises due to the presence of *dataset bias* [52], where task-irrelevant attributes are spuriously correlated with labels only in the training set. This can lead to models that rely on misleading correlations rather than learning the task-related features, resulting in biased models with poor generalization performance [62, 10].

To prevent models from learning detrimental bias, various methods are proposed to encourage models to prioritize learning task-relevant features. Recent studies enhance task-related features by first identifying bias-conflicting (unbiased) samples through loss [40, 36], gradients [1], or bias prediction techniques [35] during training, using an auxiliary biased model trained with Empirical Risk Minimization (ERM). Then, they amplify bias-conflicting samples by counteracting the bias-aligned (biased) samples through loss weighting [40] or weighted sampling [35]. The effectiveness of such methods largely depends on their precision of bias-conflicting sample detection. Specifically, there is a risk of erroneously amplifying malignant bias instead of task-relevant features when bias-aligned samples are inaccurately identified as bias-conflicting. Due to the limited detection performance of previous methods [40, 36, 1, 35], it presents a crucial challenge that remains unresolved.

---

*Equal contribution.

38th Conference on Neural Information Processing Systems (NeurIPS 2024).

In this paper, we address this challenge from a novel perspective of mislabeled sample detection. Inspired by the similarities between mislabeled samples and bias-conflicting samples, we delve into Influence Functions (IF;[27]), one of the standard methods for mislabeled sample detection [55, 57, 29], to identify bias-conflicting samples and propose a simple yet effective approach for biased models by leveraging them.

We first conduct a comprehensive analysis to explore the efficacy of Self-Influence (SI) [27], a variant of IF, in biased datasets. SI estimates how removing a specific training sample during training influences the prediction of the sample itself with the trained model (see Section. 2.2). By measuring SI, we can identify a minority sample that, if removed from the training set, increases the likelihood of incorrect predictions of itself by the trained model due to their discrepancies with the majority samples. In this context, leveraging SI to biased datasets is promising as bias-conflicting samples constitute the minority and contradict the dominant malignant bias learned by the model. However, we observe that unlike in mislabeled settings, directly applying SI to biased datasets is not as effective (Figure 1(a)-1(d)). Therefore, we investigate the differences between mislabeled samples and bias-conflicting samples and reveal the essential conditions for SI to effectively identify bias-conflicting samples. Note that we denote SI under found conditions as Bias-Conditioned Self-Influence (BCSI).

Building on our analysis, we propose a simple yet effective method for rectifying biased models through fine-tuning. We construct a small pivotal subset with a higher proportion of bias-conflicting samples using BCSI. While not perfect, this pivotal set can serve as an effective alternative to an unbiased set. Leveraging this pivotal set, we rectify a biased model through fine-tuning with only a few additional iterations. Extensive experiments demonstrate that our method can effectively rectify even after models are already debiased by recent methods.

Our contributions are threefold:

- We conduct a comprehensive analysis to explore the efficacy of SI in biased datasets and reveal the essential conditions for SI to accurately differentiate bias-conflicting samples, leading to Bias-Conditioned Self-Influence (BCSI).
- We propose a simple yet effective remedy through fine-tuning that utilizes a pivotal set constructed using BCSI to rectify biased models across varying bias severities.
- Our method is complementary to existing methods, capable of further rectifying models that have already undergone recent debiasing techniques.

## 2   Background

### 2.1   Learning from biased data

We consider a supervised learning setting with training data $D = \{z_n\}_{n=1}^N$ sampled from the data distribution $\mathbf{Z} := (X, Y)$, where the input $X$ is comprised of $X = (S, B, O)$ where $S$ is the task-related signal, $B$ is a task-irrelevant bias, and $O$ is the other task-independent feature. Also, $Y$ is the target label of the task, where the label is $y \in \{1, \ldots, C\}$. When the dataset is unbiased, ideally, a model learns to predict the target label using the task-relevant signal: $P_\theta(Y|X) = P_\theta(Y|S, B, O) = P_\theta(Y|S)$. However, when the dataset is biased, the task-irrelevant bias $B$ is highly correlated with the task-relevant features $S$ with probability $p_y$, *i.e.*, $P(B = b_y|S = s_y) = p_y$, where $p_y \geq \frac{1}{C}$. Under this relationship, a data sample $x = (s, b, o)$ is *bias-aligned* if $(b = b_y) \wedge (s = s_y)$ and, *bias-conflicting* otherwise, where $\wedge$ denotes the logical conjunction. When $B$ is easier to learn than $S$ for a model, the model may discover a shortcut solution to the given task, learning to predict $P_\theta(Y|X) = P(Y|B)$ instead of $P_\theta(Y|X) = P(Y|S)$. However, debiasing a model inclines the model towards learning the true task-signal relationship $P_\theta(Y|X) \approx P(Y|S)$.

### 2.2   Influence Functions

Influence Function (IF; [11, 27]) estimates the impact of an individual sample from the training set on the model parameters, which in turn influences model predictions. A brute-force approach to assess the influence of a sample is to exclude the data point from the training set and retrain the model to compare differences in performance, referred to as leave-one-out (LOO) retaining. However, performing LOO retraining for all samples is computationally challenging; as an alternative, influence functions have been introduced as an efficient approximated method.

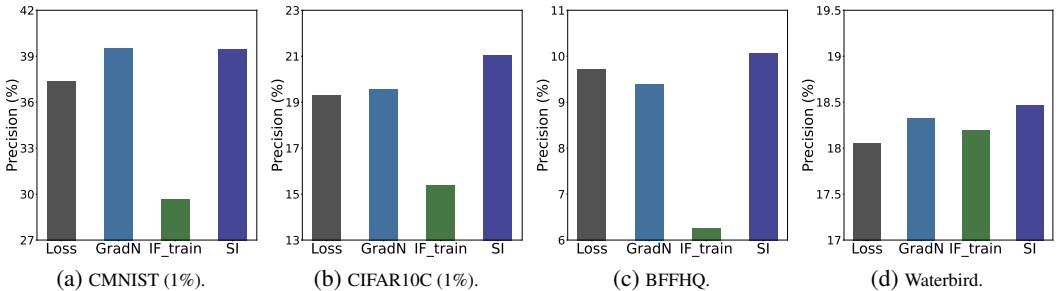

Figure 1: Precision of detecting bias-conflicting samples among Loss, Gradient Norm, Influence function on training set ($IF_{train}$), and Self-Influence (SI). The precision is evaluated with the ground truth number of bias-conflicting samples. The average precision of loss value, gradient norm, SI, and IF are presented in bars across three runs.

Here, we review the formal definition of influence function. Given a training dataset $D = \{z_n\}_{n=1}^{N}$ where $z_n = (x_n, y_n)$, model parameters $\theta$ are learned with a loss function $\mathcal{L}$:

$$\theta^* = \operatorname*{argmin}_{\theta} \mathcal{L}(D, \theta) = \operatorname*{argmin}_{\theta} \sum_{n=1}^{N} \ell(z_n, \theta)$$

where $\ell(z_n, \theta) = -\log(P_\theta(y_n|x_n))$ is the cross-entropy loss for $z_n$ with parameter $\theta$.

To measure the impact of a single training sample $z$ on model parameters $\theta$, we consider the retrained parameter $\theta_{z,\epsilon}^*$ obtained by up-weighting the loss of $z$ by $\epsilon$:

$$\theta_{z,\epsilon}^* = \operatorname*{argmin}_{\theta}(\mathcal{L}(D, \theta) + \epsilon \cdot \ell(z, \theta)).$$

Then, IF, the impact of $z$ on another sample $z'$, is defined as the deviation of the retrained loss $\ell(z', \theta_{z,\epsilon}^*)$ from the original loss $\ell(z', \theta^*)$:

$$\mathcal{I}_\epsilon(z, z') = \ell(z', \theta_{z,\epsilon}^*) - \ell(z', \theta^*)$$

For infinitesimally small $\epsilon$, we have

$$\mathcal{I}(z, z') = \left. \frac{d\mathcal{I}_\epsilon(z, z')}{d\epsilon} \right|_{\epsilon=0} = \nabla_\theta \ell(z', \theta^*)^\top H^{-1} \nabla_\theta \ell(z, \theta^*) \tag{1}$$

where $H := \nabla_\theta^2 \mathcal{L}(D, \theta^*) \in \mathbb{R}^{P \times P}$ is the Hessian of the loss function with respect to the model parameters at $\theta^*$. Intuitively, the influence $\mathcal{I}(z, z')$ estimates the effect of $z$ on $z'$ through the learning process of the model parameters. Note that IF is commonly computed once a model has converged since Equation 1 approximates more accurately when the average gradient norm of the training set is sufficiently small.

Influence function also can be calculated on itself to measure the Self-influence of $z$:

$$\mathcal{I}_{\texttt{self}}(z) \approx \nabla_\theta \ell(z, \theta^*)^\top H^{-1} \nabla_\theta \ell(z, \theta^*),$$

which approximates the difference in loss of $z$ when $z$ itself is excluded from the training set. This metric is commonly used for detecting mislabeled training samples in the noisy label setting [27, 51, 55, 57, 29] or important samples in data pruning for efficient training [49, 59].

## 3 An analysis of Self-Influence in bias-conflicting sample detection

In this section, we conduct a comprehensive analysis to delve into the efficacy of SI in bias-conflicting sample detection. First, we examine the process of identifying bias-conflicting sample detection through the perspective of mislabeled sample detection (Section 3.1). Next, we introduce essential conditions required for SI to effectively identify bias-conflicting samples by analyzing the differences between mislabeled and bias-conflicting samples (Section 3.2). We term the SI calculated under these conditions as Bias-Conditioned Self-Influence (BCSI) and demonstrate that BCSI outperforms SI in detecting bias-conflicting samples.

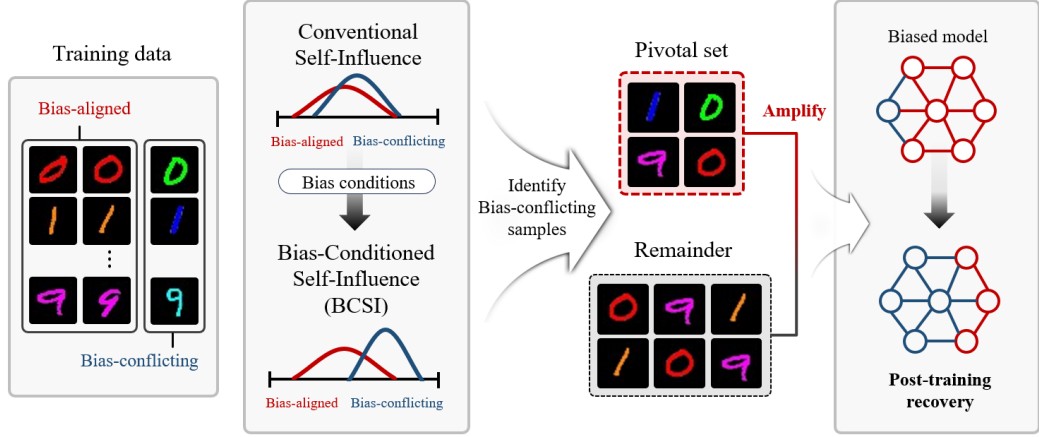

Figure 2: The overview of our method. We compute Bias-Conditioned Self-Influence (BCSI) of the training data and construct a small but concentrated pivotal set with a high ratio of bias-conflicting samples. Then, we remedy biased models through fine-tuning that utilizes the pivotal set and remaining samples.

## 3.1 Bias-conflicting sample detection from the perspective of mislabeled sample detection

IF is one of the standard methods for mislabeled sample detection [27]. The use of influence functions for mislabeled sample detection generally involves two approaches: computing influence scores using a clean validation set or computing self-influence scores. The former, $\mathcal{I}(z_i, \mathcal{V})$, utilizes a validation set $\mathcal{V}$ free of mislabeled samples to measure the impact on validation loss, identifying samples whose removal reduces this loss as likely mislabeled. The latter, Self-influence $\mathcal{I}(z_i, z_i)$, estimates how the loss of a sample $z_i$ changes when it is removed from the training set. If removing a sample significantly increases its own loss, it indicates that the sample is likely mislabeled, as normal samples can still be correctly predicted using the remaining samples. For instance, in a task classifying dogs and cats, if a dog image is mislabeled as a cat, removing this mislabeled sample from the training set decreases the likelihood of correctly predicting it as a cat.

In this context, mislabeled samples and bias-conflicting samples share a key characteristic that both are minority samples contradicting the dominant features learned by the model. Mislabeled samples have incorrect labels that conflict with the learned features, making them easily identifiable through SI. Similarly, bias-conflicting samples contradict the malignant bias that a model learns from a biased dataset. Despite the different contexts, both types of samples can be detected through the same underlying principle of IF.

In summary, given the similarities between mislabeled samples and bias-conflicting samples, it is promising to leverage the perspective and methodology of mislabeled sample detection to identify bias-conflicting samples. However, in real-world scenarios, preemptively identifying malignant bias and constructing an unbiased validation set to mitigate the bias problem is impractical. Therefore, using self-influence offers a more feasible and practical solution for addressing bias-conflicting samples instead of using influence scores on a validation set. Consequently, we center our approach on SI to effectively detect bias-conflicting samples.

## 3.2 Bias-Conditioned Self-Influence (BCSI)

To validate Self-Influence (SI) in detecting bias-conflicting samples, we conduct experiments on benchmark datasets with diverse bias types and severities: Colored MNIST, Corrupted CIFAR10, Biased FFHQ (BFFHQ), and Waterbird. These datasets feature bias related to color, synthetic corruption, gender, and place background, respectively (details in Appendix N.1). In contrast to the mislabeled setting, we observe that directly applying SI to detect bias-conflicting samples in biased datasets often fails. In Figure 1, the detection precision of SI is significantly low, mostly below 25%. Note that since an unbiased validation set is unavailable in our target problem, we additionally estimate the influence score on the training set, indicated as $IF_{train}$ in Figure 1.

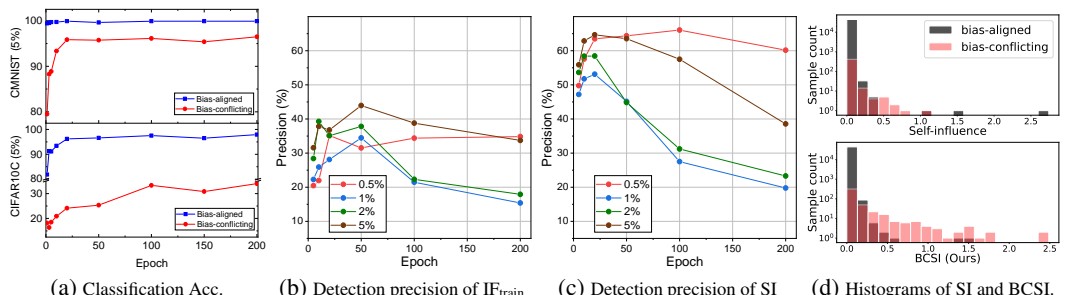

(a) Classification Acc.  (b) Detection precision of IF$_{\text{train}}$  (c) Detection precision of SI  (d) Histograms of SI and BCSI.

Figure 3: A comprehensive analysis of Influence function on the training set (IF$_{\text{train}}$) and Self-Influence (SI) in biased datasets. Figure 3(a) shows the classification accuracy of bias-aligned and bias-conflicting samples over training epochs. Figure 3(b) and 3(c) depict the detection precision of IF$_{\text{train}}$ and SI across training epochs for varying ratios of bias-conflicting samples in CIFAR10C. Figure 3(d) shows histograms of sample distribution in CIFAR10C (1%) and each bar indicates the number of samples within a specific range.

This is due to the inherent differences between mislabeled samples and bias-conflicting samples. While mislabeled samples strongly conflict with the dominant features learned by the model due to their incorrect labels, bias-conflicting samples share task-related features with bias-aligned samples. For instance, in a biased dataset where seagulls are spuriously correlated with sea backgrounds, a seagull image against a desert background still retains the features of a seagull. Despite the dominance of malignant bias, these features are still partially utilized. Therefore, bias-conflicting samples do not exhibit a clear contrast with the dominant features of a biased model, posing a challenge for using SI.

To address this challenge, we introduce essential conditions that enable SI to accurately detect bias-conflicting samples. The key concept is to restrict the model from learning task-related features and instead induce the model to focus more on the malignant bias to achieve better separation. A simple but effective way to attain this is by leveraging models in the early stages of training, since malignant bias is learned first, followed by task-related features later, according to Nam et al. [40]. In Figure 3(a), experiments on CIFAR10C and CMNIST demonstrate that the classification accuracy of bias-aligned samples increases rapidly, while that of bias-conflicting samples shows a slower rise. In addition, as shown in Figure 3(b) and 3(c), our experiments on CIFAR10C with diverse ratios of bias-conflicting samples (0.5%, 1%, 2%, and 5%) demonstrate a significant decline in detection precision of IF and SI as training epochs increase, since the model gradually learns task-related features. Therefore, computing SI with models in the early stages of training can achieve better separation. Formally, given a model parameterized by $\theta$ at an early epoch $t$, we compute the self-influence $\mathcal{I}_{\texttt{self}}(z)$ as:

$$\mathcal{I}_{\texttt{self}}(z) = \nabla_{\theta_t}\ell(z, \theta_t)^\top H_t^{-1} \nabla_{\theta_t}\ell(z, \theta_t), \qquad (2)$$

where $H_t$ is the Hessian of the loss function at the parameter $\theta_t$.

To further enhance the separation of SI, we employ Generalized Cross Entropy (GCE) [61] to induce the model to focus more on the easier-to-learn bias-aligned samples, resulting in a more biased model. GCE emphasizes samples that are easier to learn, thereby amplifying the model's bias by tending to give more weight to bias-aligned samples in the training set.

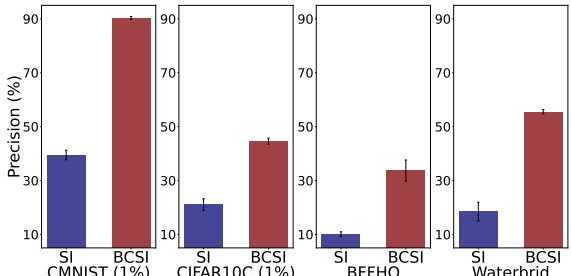

Figure 4: Comparison of average precision between SI and BCSI across diverse datasets over three runs.

Consequently, we employ the model trained under these conditions to measure SI and refer to SI estimated by this heavily biased model with Equation 2 as Bias-Conditioned Self-Influence (BCSI). Since we induce the model to heavily exploit bias and discourage the model from learning task-related features, BCSI can effectively detect bias-conflicting samples. To avoid the impracticality of manually searching epoch $t$ for each dataset, we base our method on the well-known findings of Frankle et al.

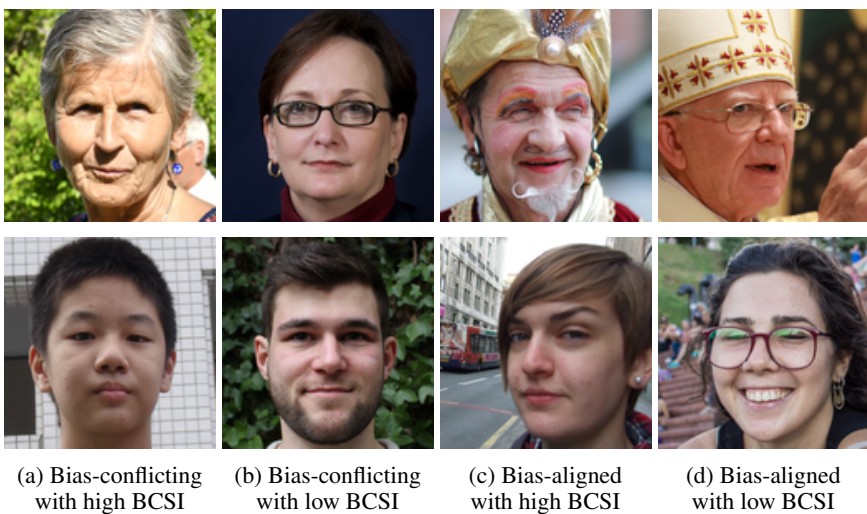

| (a) Bias-conflicting with high BCSI | (b) Bias-conflicting with low BCSI | (c) Bias-aligned with high BCSI | (d) Bias-aligned with low BCSI |

Figure 5: Example images from BFFHQ ranked within the top 100 by BCSI score. (a) and (b) are bias-conflicting samples with high and relatively lower BCSI scores, respectively. (c) is a bias-aligned sample with a high BCSI score, while (d) is a bias-aligned sample with a low BCSI score.

[9] that the primary directions of the model's parameter weights are determined during 500 to 2,000 iterations. Thus, we set epoch $t$ within this range according to the mini-batch size of each dataset. Specifically, we used $t$=5 for all datasets to ensure practicability and consistency across experiments, but fine-tuning the epoch $t$ for each dataset can yield further improvement.

We validate the efficacy of BCSI in detecting bias-conflicting samples. Since calculating $H^{-1} := (\nabla_\theta^2 L(D, \theta^*))^{-1}$ is computationally expensive for large networks due to their extensive number of parameters, we calculate $H^{-1}$ and the loss gradient of the sample $z$, $\nabla_\theta \ell(z, \theta^*)$, by using the last layer of the model following Koh and Liang [27], Pruthi et al. [43]. In Figure 4, BCSI outperforms conventional SI in detection precision.

Additionally, Figure 3(d) demonstrates that BCSI has a notable tendency for bias-conflicting samples to exhibit larger scores compared to bias-aligned samples, in contrast to SI. This trend is also observed in other biased datasets, as shown in Appendix A and C. These experimental results support that BCSI can serve as an effective indicator for identifying bias-conflicting samples. To further analyze the qualitative characteristics of bias-conflicting and bias-aligned samples within the top 100 samples ranked by BCSI, we examine BCSI on BFFHQ, as illustrated in Figure 5. In BFFHQ, gender serves as the bias attribute and age as the target attribute, leading to spurious correlations between 'young' and 'woman' as well as 'old' and 'man'. For bias-conflicting samples, Figure 5(a) shows that BCSI assigns high scores to clear counterexamples, such as boys or very elderly women. In contrast, Figure 5(b) exhibits relatively lower BCSI scores for cases like slightly older young men or elderly women who appear younger, indicating that BCSI prioritizes samples with stronger opposition to spurious correlations. A similar trend is observed for bias-aligned samples in Figure 5(c) and Figure 5(d), enhancing that BCSI effectively distinguishes between varying degrees of alignment with the spurious correlations.

## 4 Remedy biased models through fine-tuning

In this section, we propose a simple but effective remedy that first utilizes BCSI to construct a concentrated pivotal subset abundant in bias-conflicting samples and then employs it for rectifying biased models via fine-tuning without leveraging the supervision of bias or an unbiased validation set. Our method is complementary to existing methods, capable of rectifying models that have already undergone other debiasing techniques. The overall pipeline is described in Figure 2.

**Constructing a pivotal subset.** We select the top-$k$ subset of samples from each class, based on their BCSI scores, to form a pivotal subset of bias-conflicting samples as follows: $\mathbf{Z}_\mathrm{P} =$

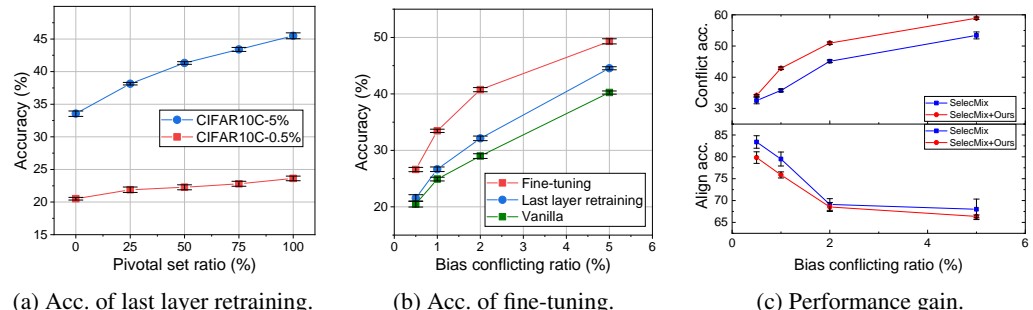

(a) Acc. of last layer retraining.     (b) Acc. of fine-tuning.     (c) Performance gain.

Figure 6: Test accuracy under varying bias-conflicting ratios. Figure 6(a) shows the accuracy for last layer retraining across varying bias ratios in pivotal sets. Figure 6(b) depicts performance changes of last layer retraining and fine-tuning under diverse bias ratios. In Figure 6(c), our performance gains are provided. We present the average accuracy with the error bars indicating the standard error across three runs.

$\bigcup_{c=1}^{C} \left\{ z_{\text{BCSI-rank}(n,c)} \right\}_{n=1}^{k}$, where $C$ is the number of classes and $\text{BCSI-rank}(n, c)$ is the dataset index of the $n$-th training sample of class $c$ sorted by BCSI score. Due to the unknown ratio of bias-conflicting samples beforehand, determining a proper $k$ through hyper-parameter tuning for each dataset is computationally expensive. To mitigate this, we repeat the selection process three times with different randomly initialized models and use the intersection of the resulting sets as the pivotal set. This ensures a high likelihood of selecting bias-conflicting samples, as they are consistently identified by three runs. We provide the resulting bias-conflicting ratios of the pivotal sets across various datasets in Appendix E and confirm that this process improves detection precision by 16.11% on average. Note that we set $k = 100$ across all datasets in our experiments. For computational costs, since we only train models for five epochs, this iterative approach incurs negligible cost compared to full training, as commonly done in previous studies [40, 31, 19]. We confirm that in Appendix G. The detailed filtering process is outlined in Algorithm 1, which can be found in Appendix B.

**Efficient remedy via fine-tuning.**   Recent works [26, 32] show that retraining specific layers of a model using a small unbiased set can effectively mitigate bias in biased models, overcoming the inefficiency of retraining models from scratch [40, 31, 19]. However, preemptively identifying the bias and curating an unbiased set is very costly, making it an impractical solution. Instead, our method leverages the pivotal set which has a high proportion of bias-conflicting samples, as a practical alternative. While not perfect, our method can efficiently remedy biased models with just a few additional training iterations without the need for prior knowledge of bias or unbiased datasets. As shown in Figure 6(a), even without a perfect pivotal set, its concentration facilitates its applicability in fine-tuning. Note that contrary to the claims of Kirichenko et al. [26], we observed that in highly biased scenarios, the feature extractor also becomes biased, making last-layer retraining insufficient, as demonstrated in Figure 6(b). Therefore, we fine-tune all the parameters in the models.

In addition, we formulate a counterweight cross-entropy loss by drawing a mini-batch from the remaining training set. In real-world scenarios, the unpredictability of bias severity necessitates robustness across a wide range of bias severities. However, previous methods often assume a sufficient presence of bias-aligned samples in the training set, which limits their performance in low-bias scenarios. Despite its significance, the study on both low and high-bias scenarios has been underexplored, and to the best of our knowledge, we are the first to bring up this issue.

We then train the model using both the cross-entropy loss on the pivotal subset and the counterweight loss on the remaining training set:

$$\mathcal{L}(\mathbf{Z}_{\text{P}}, \mathbf{Z}_{\text{R}}) := \mathcal{L}_{\text{CE}}(\mathbf{Z}_{\text{P}}) + \lambda \mathcal{L}_{\text{CE}}(\mathbf{Z}_{\text{R}}),$$

where $\mathbf{Z}_{\text{P}}$ is the pivotal subset, $\mathbf{Z}_{\text{R}} \sim \mathbf{Z} \setminus \mathbf{Z}_{\text{P}}$ a randomly drawn mini-batch from the remaining training set, and $\mathcal{L}_{\text{CE}}$ is the mean cross-entropy loss.

To this end, our method efficiently remedies bias through fine-tuning that utilizes a pivotal set constructed via BCSI across varying bias severities. Additionally, our approach complements existing methods, capable of further rectifying models that have already undergone recent debiasing techniques. The overall process is described in Algorithm 2, which is included in Appendix B.

Table 1: The average and the standard error over three runs. *Ours* indicates our method applied to a model initially trained with the prefix method. The best accuracy is annotated in **bold**. ✓ indicates that a given method uses bias information while ✗ denotes that a given model does not use any bias information.

| Method | Bias Info | CMNIST | | | | CIFAR10C | | | | BFFHQ |
|---|---|---|---|---|---|---|---|---|---|---|
| | | 0.5% | 1% | 2% | 5% | 0.5% | 1% | 2% | 5% | 0.5% |
| GroupDRO | ✓ | 79.57 | 90.50 | 94.89 | 97.54 | 33.44 | 38.30 | 45.81 | 57.32 | 54.80 |
| ERM | ✗ | $71.76_{\pm1.84}$ | $86.47_{\pm0.61}$ | $93.87_{\pm0.32}$ | $96.28_{\pm0.29}$ | $20.50_{\pm0.54}$ | $24.91_{\pm0.33}$ | $28.99_{\pm0.42}$ | $40.24_{\pm0.28}$ | $53.53_{\pm2.05}$ |
| LfF | ✗ | $89.06_{\pm1.87}$ | $89.50_{\pm2.88}$ | $85.74_{\pm4.37}$ | $94.30_{\pm0.67}$ | $25.28_{\pm2.89}$ | $31.15_{\pm1.67}$ | $38.64_{\pm0.39}$ | $46.15_{\pm0.54}$ | $55.33_{\pm2.69}$ |
| DFA | ✗ | $84.71_{\pm1.66}$ | $90.20_{\pm1.29}$ | $92.31_{\pm0.77}$ | $94.33_{\pm1.23}$ | $27.13_{\pm1.66}$ | $31.26_{\pm2.71}$ | $37.96_{\pm0.71}$ | $44.99_{\pm0.84}$ | $52.07_{\pm1.91}$ |
| BPA | ✗ | $73.34_{\pm2.37}$ | $87.21_{\pm0.30}$ | $89.42_{\pm3.37}$ | $97.13_{\pm0.15}$ | $25.50_{\pm1.03}$ | $26.86_{\pm0.69}$ | $27.47_{\pm1.46}$ | $34.29_{\pm2.20}$ | $51.40_{\pm2.98}$ |
| DCWP | ✗ | $85.16_{\pm7.75}$ | $89.68_{\pm6.95}$ | $89.42_{\pm4.23}$ | $95.17_{\pm1.75}$ | $31.27_{\pm0.24}$ | $34.87_{\pm0.63}$ | $41.47_{\pm0.06}$ | $52.86_{\pm1.24}$ | $57.33_{\pm1.75}$ |
| SelecMix | ✗ | $84.46_{\pm0.58}$ | $94.51_{\pm0.53}$ | $95.75_{\pm1.34}$ | $98.09_{\pm0.13}$ | $37.63_{\pm0.81}$ | $40.14_{\pm0.42}$ | $47.54_{\pm0.59}$ | $54.86_{\pm0.76}$ | $63.07_{\pm2.32}$ |
| Ours ERM | ✗ | $75.87_{\pm1.60}$ | $89.69_{\pm0.41}$ | $95.08_{\pm0.17}$ | $96.79_{\pm0.13}$ | $26.61_{\pm0.38}$ | $33.47_{\pm0.29}$ | $40.75_{\pm0.37}$ | $49.30_{\pm0.46}$ | $56.00_{\pm1.07}$ |
| Ours LfF | ✗ | $\mathbf{90.79}_{\pm1.13}$ | $94.10_{\pm1.08}$ | $92.95_{\pm1.17}$ | $95.59_{\pm0.53}$ | $27.63_{\pm1.00}$ | $35.29_{\pm1.21}$ | $43.36_{\pm0.78}$ | $51.95_{\pm0.29}$ | $57.13_{\pm2.46}$ |
| Ours DFA | ✗ | $88.39_{\pm0.28}$ | $92.85_{\pm067}$ | $95.67_{\pm0.12}$ | $97.52_{\pm0.06}$ | $25.66_{\pm0.85}$ | $33.53_{\pm2.01}$ | $42.80_{\pm0.81}$ | $52.61_{\pm0.54}$ | $56.60_{\pm2.83}$ |
| Ours SelecMix | ✗ | $87.63_{\pm1.20}$ | $\mathbf{95.35}_{\pm0.17}$ | $\mathbf{97.15}_{\pm0.48}$ | $\mathbf{98.13}_{\pm0.17}$ | $\mathbf{38.74}_{\pm0.36}$ | $\mathbf{46.18}_{\pm0.33}$ | $\mathbf{52.70}_{\pm0.40}$ | $\mathbf{59.66}_{\pm0.31}$ | $\mathbf{65.80}_{\pm3.12}$ |

# 5 Experiments

In this section, we present experiments applying our method to models trained with ERM and recent debiasing methods. We validate our method and its individual components by following prior conventions. Below, we provide a brief overview of our experimental setting in Section 5.1, followed by empirical results presented in Section 5.2, 5.3, and 5.4.

## 5.1 Experimental settings

We now describe datasets, baselines, and evaluation protocol. Detailed descriptions about these are provided in Appendix N.

**Datasets.** For a fair evaluation, we follow the conventions of using benchmark biased datasets [40]. Colored MNIST dataset (CMNIST) is a synthetically modified MNIST [6], where the labels are correlated with colors. We conduct benchmarks on bias ratios of $r \in \{0.5, 0.1, 0.2, 5\}$. CIFAR10C is a synthetically modified CIFAR10 [30] dataset with common corruptions. To evaluate our method in low-bias scenarios, we expand our scope and conduct experiments with varying bias ratios $r \in \{0.5, 0.1, 0.2, 5, 20, 30, 50, 70, 90(\text{unbiased})\}$. Biased FFHQ (BFFHQ) [31] is a curated Flickr-Faces-HQ (FFHQ) [22] dataset, which consists of facial images where ages and genders exhibit spurious correlation. The Waterbirds dataset [53] consists of bird images, to classify bird types, but their backgrounds are correlated with bird types. Non-I.I.D. Image dataset with Contexts (NICO) [16] is a natural image dataset for out-of-distribution classification. We follow the setting of [54], inducing long-tailed bias proportions within each class, simulating diverse bias ratios in a single benchmark. Additionally, to demonstrate the effectiveness of our method on NLP datasets, we conduct experiments on CivilComments [2, 28] and MultiNLI [58, 45], as detailed in appendix F.

**Baselines.** Since our goal is addressing the dataset bias without leveraging any prior knowledge of bias or an unbiased set, we evaluate our method with such baselines. GroupDRO [44] uses bias supervision to debias models. LfF [40], BPA [48], and DCWP [41] adjust the loss function to amplify the learning signals for bias-conflicting samples. DFA [31] and SelecMix [19] augment samples possessing various biases different from the original data.

**Evaluation protocol.** Following other baselines, we calculate the accuracy for unbiased test sets in CMNIST, CIFAR10C, and NICO. We measure the minority-group accuracy in BFFHQ, and the worst-group accuracy in Waterbird. Note that we use the models from the final epoch for all

Table 2: The average and the standard error over three runs on low-bias scenarios.

| Method | CIFAR10C | | | | |
|---|---|---|---|---|---|
| | 20% | 30% | 50% | 70% | 90%(unbiased) |
| ERM | $59.47_{\pm0.59}$ | $65.64_{\pm0.51}$ | $71.33_{\pm0.09}$ | $\mathbf{74.90}_{\pm0.25}$ | $\mathbf{76.03}_{\pm0.26}$ |
| LfF | $59.78_{\pm0.85}$ | $60.56_{\pm0.96}$ | $60.35_{\pm0.37}$ | $62.52_{\pm0.49}$ | $63.42_{\pm0.63}$ |
| DFA | $60.34_{\pm0.46}$ | $64.24_{\pm0.44}$ | $65.97_{\pm1.80}$ | $64.97_{\pm0.20}$ | $66.59_{\pm5.20}$ |
| SelecMix | $62.05_{\pm1.26}$ | $62.17_{\pm0.35}$ | $62.52_{\pm1.54}$ | $66.23_{\pm0.09}$ | $65.81_{\pm0.96}$ |
| Ours ERM | $62.78_{\pm0.67}$ | $65.61_{\pm0.77}$ | $70.61_{\pm0.62}$ | $73.20_{\pm0.35}$ | $73.57_{\pm0.16}$ |
| Ours LfF | $64.46_{\pm0.29}$ | $64.40_{\pm0.27}$ | $65.82_{\pm0.15}$ | $67.29_{\pm0.17}$ | $68.15_{\pm0.76}$ |
| Ours DFA | $66.30_{\pm0.48}$ | $\mathbf{68.13}_{\pm0.45}$ | $\mathbf{72.79}_{\pm0.38}$ | $73.56_{\pm0.15}$ | $70.36_{\pm4.08}$ |
| Ours SelecMix | $\mathbf{66.67}_{\pm0.43}$ | $64.51_{\pm1.44}$ | $66.45_{\pm0.28}$ | $69.97_{\pm0.21}$ | $69.29_{\pm0.75}$ |

Table 3: Accuracy on Waterbirds, NICO

| Method | Waterbird | NICO |
|---|---|---|
| ERM | $68.74_{\pm2.65}$ | $39.56_{\pm1.77}$ |
| LfF | $75.27_{\pm2.12}$ | $34.56_{\pm1.47}$ |
| DFA | $77.57_{\pm1.60}$ | $44.59_{\pm0.33}$ |
| SelecMix | $74.72_{\pm1.14}$ | $33.87_{\pm1.27}$ |
| Ours ERM | $87.64_{\pm1.30}$ | $43.54_{\pm0.50}$ |
| Ours LfF | $87.85_{\pm0.68}$ | $40.18_{\pm0.91}$ |
| Ours DFA | $87.12_{\pm0.68}$ | $\mathbf{45.69}_{\pm1.12}$ |
| Ours SelecMix | $\mathbf{89.67}_{\pm0.38}$ | $44.33_{\pm0.55}$ |

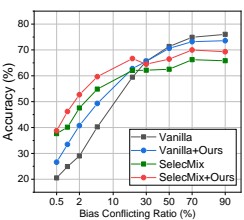

(a) Acc. on bias ratios.

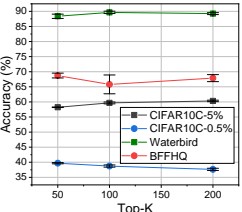

(b) Acc. on varying $k$.

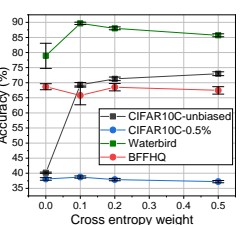

(c) Acc. on varying $\lambda$.

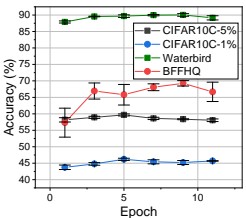

(d) Acc. on epochs.

Figure 7: Figure 7(a) displays the test accuracy for SelecMix and our method at different bias ratios. Figure 7(b), 7(c), and 7(d) depict the unbiased evaluation under varying the size $k$ in the pivotal set, $\lambda$, the number of epochs for detection models, respectively. We present the average accuracy with the error bars indicating the standard error across three runs.

experiments to evaluate performance. We report the average value and the error bars denote standard errors across three runs.

## 5.2 Results on highly biased scenarios

We evaluate our method to measure the degree of rectification of baseline models when combined with ours on benchmark datasets. In Table 1, we significantly enhance the performance of baselines on the majority of datasets under various experimental settings. Ours SelecMix achieves state-of-the-art accuracy on CIFAR10C. Also, we observe that performance gain is larger as the ratio of bias-conflicting samples increases in CIFAR10C. We conjecture that fine-tuning becomes more effective in CIFAR10C (2%) and (5%) since the bias-conflicting sample purity of the pivotal set increases, as shown in Section 4. In Figure 6(c), performance gain mainly stems from the increased performance of bias-conflicting samples.

## 5.3 Results on low-bias scenarios

We validate the baselines on CIFAR10C under varying ratios of bias-conflicting samples in Table 2 and 3. Baselines exhibit performance deterioration compared to ERM when the bias-conflicting ratio is high. In contrast, our method can significantly rectify remaining bias within a model, even in mildly biased datasets except for ERM. Albeit there is a slight decrease in performance for ERM, the accuracy gap is much lower than other baselines. Since the innate nature of fine-tuning can minimize friction by training from pre-trained parameters, our method can remedy bias in a wider range of bias ratios, as in Figure 7(a). The results for other methods are provided in Appendix D.

## 5.4 Ablation study

We examine the sensitivity of hyperparameters such as the number of selected samples per class ($k$) in the pivotal set, the weight for the remaining data in fine-tuning ($\lambda$), and the number of epochs

to train detection models. In Figure 7(b), there is a slight performance decrease as $k$ increases in CIFAR10C (0.5%). In contrast, the accuracy in CIFAR10C (5%) increases. Since there are a few bias-conflicting samples per class in CIFAR10C (0.5%), additional usage of samples dilutes the ratio of bias-conflicting data in the pivotal set, leading to a performance drop. In Figure 7(c), we observe a marginal accuracy drop as $\lambda$ increases in CIFAR10C (0.5%), CIFAR10C (90%) experiences a performance increase. These results indicate that learning the remaining samples is beneficial in CIFAR10C (90%), fostering the model to capture task-relevant signals. For the number of epochs used to train the model for the detection, we compute the final performance when combining SelecMix in Figure 7(d). Except for insufficiently trained 1 epoch, the performance is not sensitive to the number of epochs between 3 and 11 epochs. We note that the analysis for intersections is provided in Appendix H.

## 6 Related work

**Debiasing deep neural networks.** Research on mitigating bias has centered on modulating task-related information and malignant bias during training. Early works relied on human knowledge through direct supervision or implicit information of bias [44, 33, 18, 12], which is often impractical due to its acquisition cost. Thus, several studies have focused on identifying and utilizing bias-conflicting samples without relying on human knowledge. Loss modification methods [40, 36, 41] amplify the learning signals of (estimated) bias-conflicting samples by modifying the learning objective. Sampling methods [35, 1] overcome dataset bias by sampling (estimated) bias-conflicting data more frequently. Data augmentation approaches [31, 34, 20, 19] synthesize samples with various biases distinct from the inherent bias of the original data. Recently, based on the observation that bias in classification layers is severe compared to feature extractors, several approaches focus on rectifying the last layer [23, 39, 26]. Similarly, Lee et al. [32] demonstrated that selectively fine-tuning a subset of the layers with an unbiased dataset can match or even surpass the performance of commonly used fine-tuning methods. However, identifying the bias and curating an unbiased set is very costly, making it an impractical essential condition.

**Influence functions.** Influence function (IF) and its approximations [43, 46, 24] have been utilized in various deep learning tasks by measuring the importance of training samples and the relationship between them. One application of IF is in quantifying memorization by self-influence, which is the increase in loss when a training sample is excluded [43, 8]. IF can be used to estimate the significance of samples, enabling the reduction of less important ones for efficient training [49, 59]. Recent works utilize IF to identify and relabel mislabeled samples in noisy label settings [27, 51, 55, 57, 29]. Furthermore, IF has also been applied in 3D domains like NeRF, where it measures pixel-wise distraction caused by unexpected objects, aiding in the identification and mitigation of such distractions [21].

## 7 Conclusion

In this work, we introduce a novel perspective of mislabeled sample detection on biased datasets. By conducting a comprehensive analysis of Self-Influence in detecting bias-conflicting samples, we discover essential conditions required for SI to effectively identify these samples, which we denote as Bias-Conditioned Self-Influence (BCSI). Building on our analysis, we propose a simple yet effective remedy for biased models through fine-tuning that utilizes a small but concentrated pivotal set constructed via BCSI. Our method is not only capable of further rectifying models that have already undergone recent debiasing techniques but also demonstrates better generalization on a wide range of bias severities compared to previous studies.

**Limitations.** In this work, we rectify biased models via a simple fine-tuning approach. However, this is the basic method; more sophisticated techniques such as sample weighting or curriculum learning are possible. We believe that our introduction of this novel perspective will pave the way for more advanced future work.

**Broader impact.** Our work aims to learn unbiased deep learning models without bias annotations. Since filtering every training data under every given circumstance, the social impact of the ability to debias a biased deep learning model after its training is much needed in terms of fairness.

## Acknowledgements

This work was partly supported by Institute of Information & communications Technology Planning & Evaluation (IITP) grants (No.2022-0-00713 Meta-learning applicable to real-world problems, No.2022-0-00984 Development of Artificial Intelligence Technology for Personalized Plug-and-Play Explanation and Verification of Explanation, No.RS-2019-II190075 Artificial Intelligence Graduate School Program(KAIST)), National Research Foundation of Korea (NRF) grants (RS-2023-00209060 A Study on Optimization and Network Interpretation Method for Large-Scale Machine Learning) funded by the Korea government (MSIT), and KAIST-NAVER Hypercreative AI Center.

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

# A Distribution of Self-Influence and bias-conditioned Influence

In Figure 1(c) and Figure 1(d) of the main paper, we have shown the influence histogram of naive self-influence and bias-conditioned self-influence (Ours) for the training set of CIFAR10C (1%). In this section, we show the histograms of self-influence and bias-conditioned self-influence for the training sets of an extended variety of bias ratios and datasets. Figure 8 shows the influence histograms for CIFAR10C, and BFFHQ. Figure 9 shows the influence histograms of CMNIST, Waterbird, and NICO. In accordance with the main paper, we observe that bias-conditioned self-influence generally exhibits better separation compared to naive self-influence, deeming it a better option to detect bias-conflicting samples.

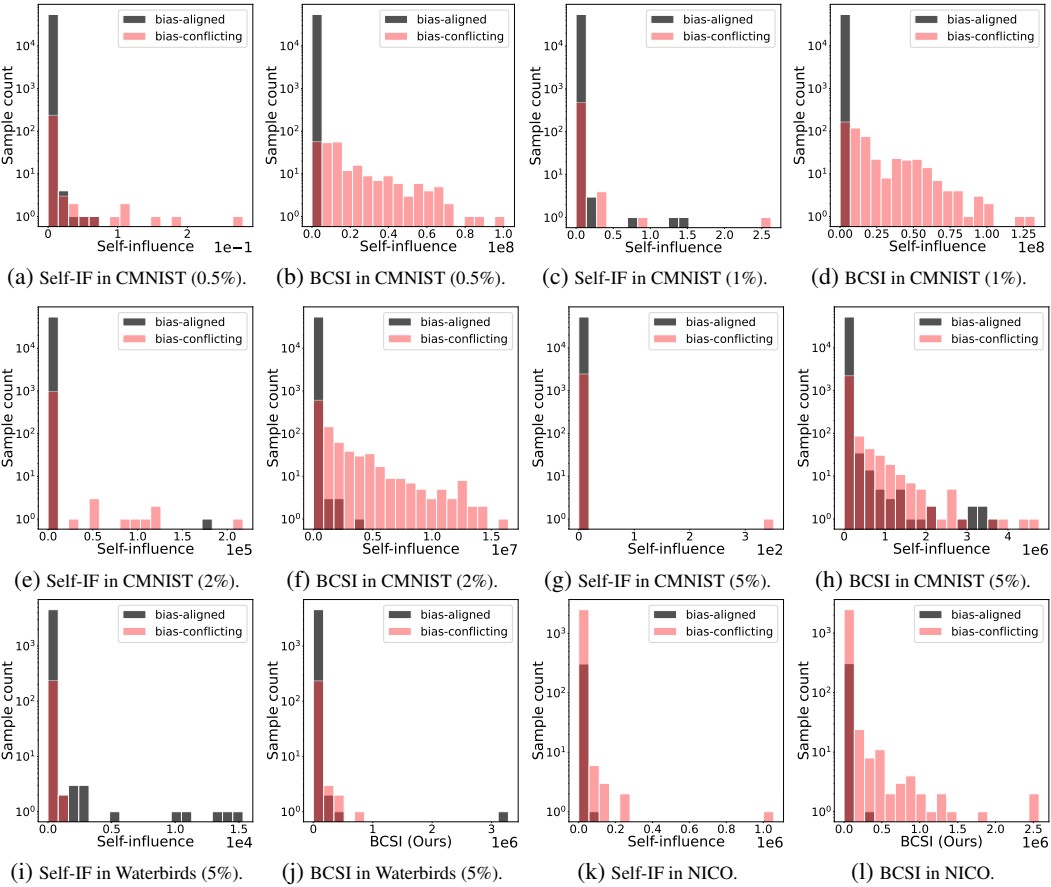

(a) Self-IF in CMNIST (0.5%).    (b) BCSI in CMNIST (0.5%).    (c) Self-IF in CMNIST (1%).    (d) BCSI in CMNIST (1%).

(e) Self-IF in CMNIST (2%).    (f) BCSI in CMNIST (2%).    (g) Self-IF in CMNIST (5%).    (h) BCSI in CMNIST (5%).

(i) Self-IF in Waterbirds (5%).    (j) BCSI in Waterbirds (5%).    (k) Self-IF in NICO.    (l) BCSI in NICO.

Figure 8: Histogram of self-influence and bias-conditioned self-influence for CMNIST, Waterbird, and NICO.

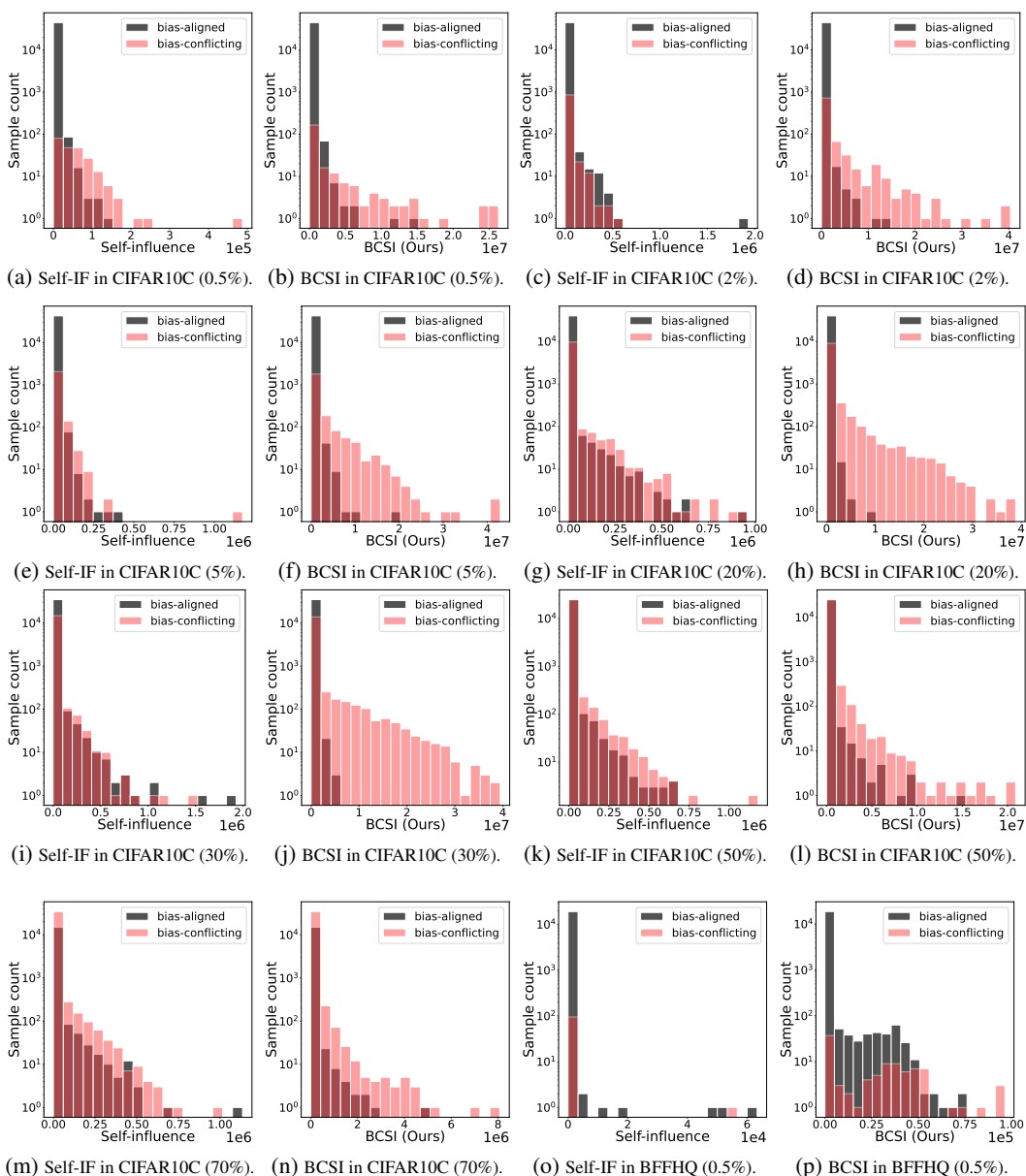

Figure 9: Histogram of self-influence and bias-conditioned self-influence for the CIFAR10C dataset with varying bias-conflicting ratio and BFFHQ.

(a) Self-IF in CIFAR10C (0.5%). (b) BCSI in CIFAR10C (0.5%). (c) Self-IF in CIFAR10C (2%). (d) BCSI in CIFAR10C (2%).

(e) Self-IF in CIFAR10C (5%). (f) BCSI in CIFAR10C (5%). (g) Self-IF in CIFAR10C (20%). (h) BCSI in CIFAR10C (20%).

(i) Self-IF in CIFAR10C (30%). (j) BCSI in CIFAR10C (30%). (k) Self-IF in CIFAR10C (50%). (l) BCSI in CIFAR10C (50%).

(m) Self-IF in CIFAR10C (70%). (n) BCSI in CIFAR10C (70%). (o) Self-IF in BFFHQ (0.5%). (p) BCSI in BFFHQ (0.5%).

# B Algorithm

**Algorithm 1** Construct a pivotal set

1: **Input:** model parameters $\theta$, GCE $\mathcal{L}_{\text{GCE}}$, number of epochs $n_{\text{epoch}}$, learning rate $\rho$, number of classes $C$, train set $\mathbf{Z}$, number of topk $n_{\text{topk}}$
2: **Initialize:** Model parameter $\theta$.
3: **for** $i = 0, 1, 2, \cdots, n_{\text{epoch}}$ **do**
4:     $\theta \leftarrow \theta - \rho \nabla_\theta \mathcal{L}_{\text{GCE}}(\mathbf{Z}, \theta)$
5: **end for**
6: # Select samples with high self-influence
7: $\mathbf{Z}_{\text{P}} \leftarrow \emptyset$
8: **for** $c = 0, 1, 2, \cdots, C$ **do**
9:     $\mathbf{Z}_c \leftarrow \{(x, y) \in \mathbf{Z} | y = c\}$
10:     **for** $j = 0, 1, 2, \cdots, n_{\text{topk}}$ **do**
11:       $z_{\text{highest}} \leftarrow \text{argmax}_{z \in \mathbf{Z}_c} \mathcal{I}_{\text{self}}(z)$
12:       $\mathbf{Z}_c \leftarrow \mathbf{Z}_c \setminus \{z_{\text{highest}}\}$
13:       $\mathbf{Z}_{\text{P}} \leftarrow \mathbf{Z}_{\text{P}} \cup \{z_{\text{highest}}\}$
14:     **end for**
15: **end for**
16: **Output:** $\mathbf{Z}_{\text{P}}$

**Algorithm 2** Post-training with the pivotal set

1: **Input:** pre-trained model parameters $\theta^*$, CE $\mathcal{L}_{\text{CE}}$, number of iterations $n_{\text{iter}}$, learning rate $\rho$, train set $\mathbf{Z}$, pivotal set $\mathbf{Z}_{\text{P}}$, weight of remaining set $\lambda$
2: **Initialize:** Last-layer of model $\theta^*_{\text{last-layer}}$.
3: $\mathbf{Z}_{\text{R}} \leftarrow \mathbf{Z} \setminus \mathbf{Z}_{\text{P}}$
4: $n_{\text{P}} \leftarrow |\mathbf{Z}_{\text{P}}|$
5: **for** $i = 0, 1, 2, \cdots, n_{\text{iter}}$ **do**
6:     # Sample data from remaining samples
7:     $\mathbf{Z}_{\text{S}} \leftarrow \emptyset$
8:     **for** $j = 0, 1, 2, \cdots, n_{\text{P}}$ **do**
9:       $z \sim \mathbf{Z}_{\text{R}}$
10:       $\mathbf{Z}_{\text{S}} \leftarrow \mathbf{Z}_{\text{S}} \cup \{z\}$
11:     **end for**
12:     $\mathcal{L} \leftarrow \mathcal{L}_{\text{CE}}(\mathbf{Z}_{\text{P}}, \theta^*)$
13:     $\mathcal{L} \leftarrow \mathcal{L} + \lambda \mathcal{L}_{\text{CE}}(\mathbf{Z}_{\text{S}}, \theta^*)$
14:     $\theta^* \leftarrow \theta^* - \rho \nabla_\theta \mathcal{L}$
15: **end for**
16: **Output:** $\theta^*$

# C Detection precision for other datasets

We now describe the detailed experimental setting used in Figure 4 of the main paper. We first train ResNet18 [15] for five epochs and then compute self-influence, and bias-conditioned self-influence. Note that we only use the last layer when computing self-influence, and bias-conditioned self-influence. Subsequently, we sort the training data in descending order based on the values obtained by each method, selecting samples ranging from the highest to the $k$-th sample, where $k$ is the number of total bias-conflicting samples in the training set. We then calculate the precision in detecting bias-conflicting samples within the selected data.

To further demonstrate the effectiveness of bias-conditioned self-influence in detecting bias-conflicting samples, we compare bias-conditioned self-influence with self-influence on other datasets including CMNIST (0.5%, 2%, 5%), CIFAR10C (0.5%, 2%, 5%, 20%, 50%), NICO. As shown in Table 4, bias-conditioned self-influence exhibits superior performance or is comparable to self-influence in most cases. This observation is consistent with the result in the main paper.

Table 4: Comparison of bias-conflicting sample detection precisions between self-influence (SI), and bias-conditioned self-influence (BCSI) across various datasets. The average and the standard error of precision over three runs are provided.

| Method | CMNIST | | | CIFAR10C | | | | | NICO |
|---|---|---|---|---|---|---|---|---|---|
| | 0.5% | 2% | 5% | 0.5% | 2% | 5% | 20% | 50% | |
| SI | $31.94_{\pm 2.85}$ | $39.35_{\pm 1.38}$ | $37.91_{\pm 0.84}$ | $\mathbf{63.33}_{\pm 1.75}$ | $20.19_{\pm 2.22}$ | $42.17_{\pm 1.74}$ | $41.11_{\pm 0.08}$ | $58.73_{\pm 0.30}$ | $89.37_{\pm 0.21}$ |
| BCSI | $\mathbf{92.08}_{\pm 0.24}$ | $\mathbf{82.86}_{\pm 1.38}$ | $\mathbf{44.00}_{\pm 3.77}$ | $38.33_{\pm 2.52}$ | $\mathbf{50.45}_{\pm 0.34}$ | $\mathbf{60.48}_{\pm 1.91}$ | $\mathbf{69.48}_{\pm 0.39}$ | $\mathbf{71.78}_{\pm 0.29}$ | $\mathbf{90.86}_{\pm 0.39}$ |

# D   Performance with respect to the bias-conflicting ratio

In Figure 4.2 of the main paper, we showed the unbiased accuracy trends of the CIFAR10C dataset with respect to the bias-conflicting ratio for SelecMix and SelecMix with our method. In Figure 10, we provide the CIFAR10C accuracy trends of LfF [40] and DFA [31] alone and with our method.

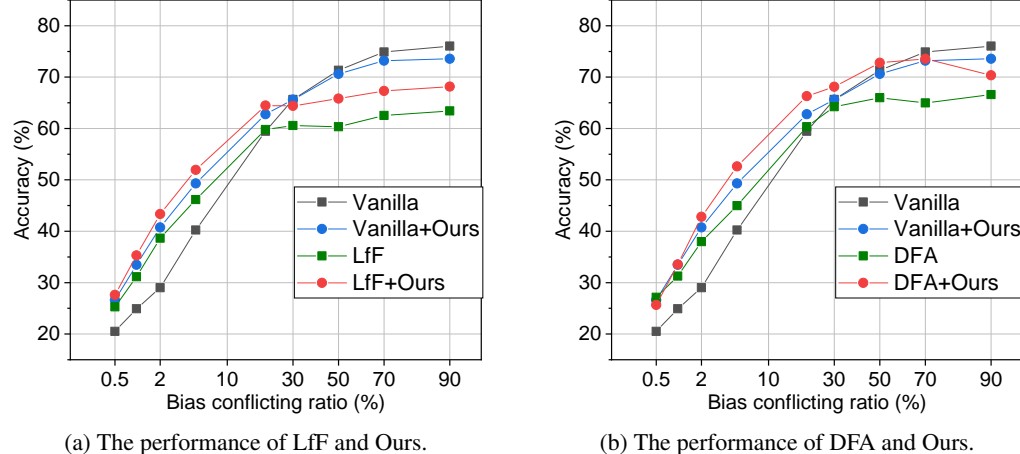

(a) The performance of LfF and Ours.          (b) The performance of DFA and Ours.

Figure 10: Performance of the other baselines and Ours on the CIFAR10C dataset with varying bias ratio. The performance of LfF [40] is shown in Figure 10(a). Figure 10(b) displays the performance of DFA [31].

# E   Bias-conflicting ratio of the pivotal set

We provide the resulting bias-conflicting ratios (*i.e.* bias-conflicting detection precisions) of the pivotal set produced across a variety of datasets. Table 5 and Table 6 show the bias-conflicting ratios for CMNIST (0.5%, 1%, 2%, 5%), CIFAR10C (0.5%, 1%, 2%, 5%, 20%, 30%, 50%, 70%), BFFHQ, Waterbirds, and NICO.

Table 5: The average and the standard error of detection precision over three runs. Note that we compute the precision of the pivotal sets across varying ratios of bias-conflicting samples in CIFAR10C.

| | CIFAR10C | | | | | | | |
| | 0.5% | 1% | 2% | 5% | 20% | 30% | 50% | 70% |
| --- | --- | --- | --- | --- | --- | --- | --- | --- |
| Accuracy | $45.57_{\pm 1.63}$ | $68.18_{\pm 0.96}$ | $86.13_{\pm 1.18}$ | $96.60_{\pm 0.11}$ | $99.94_{\pm 0.06}$ | $99.88_{\pm 0.12}$ | $98.30_{\pm 0.30}$ | $85.81_{\pm 2.05}$ |

Table 6: The average and the standard error of detection precision over three runs. Note that we compute the precision of the pivotal sets on CMNIST, BFFHQ, Waterbirds, and NICO.

| | CMNIST | | | | BFFHQ | Waterbirds | NICO |
| | 0.5% | 1% | 2% | 5% | 0.5% | 5% | |
| --- | --- | --- | --- | --- | --- | --- | --- |
| Accuracy | $68.19_{\pm 4.57}$ | $84.61_{\pm 1.86}$ | $97.27_{\pm 0.51}$ | $73.24_{\pm 6.55}$ | $66.32_{\pm 3.94}$ | $64.26_{\pm 1.93}$ | $94.01_{\pm 2.57}$ |

# F   Results on natural language processing datasets

To verify the effectiveness of our method on NLP datasets, we conduct experiments on two widely used benchmarks, CivilComments and MultiNLI. CivilComments contains user-generated comments labeled as either toxic or non-toxic. The spurious attribute in this dataset indicates whether a comment mentions one of the protected attributes, such as male, female, LGBT, black, white, Christian, Muslim, or other religions. These attributes are disproportionately associated with toxic comments, creating

a spurious correlation. Similarly, the MultiNLI dataset comprises pairs of sentences with labels denoting their relationship as contradiction, entailment, or neutral. The spurious attribute in MultiNLI is the presence of negation words, which are more frequently observed in the contradiction class. Both datasets are structured into groups based on combinations of the label $y$ and the spurious attribute $s$, resulting in four groups in *CivilComments* and six in *MultiNLI*.

As shown in table 7, our method demonstrates its effectiveness in further rectifying models previously debiased with JTT, achieving increases in worst-group accuracy of 3.4% and 14.8% on MultiNLI and CivilComments, respectively.

Table 7: The average and the worst-group accuracy on NLP datasets.

| **Method** | MultiNLI | | CivilComments | |
|---|---|---|---|---|
| | Avg. | Worst-group | Avg. | Worst-group |
| ERM | 82.4 | 67.9 | 92.6 | 57.4 |
| JTT | 80.0 | 70.2 | 92.6 | 63.7 |
| Ours JTT | 79.8 | **73.6** | 86.9 | **78.5** |

## G Comparison of time costs

In this section, we analyze the time cost of our method and compare it with the other baselines. For a practical and tangible comparison, we measure the wall clock time for the CIFAR10C (0.5%) dataset. We run our experiments with a machine equipped with Intel Xeon Gold 5215 (Cascade Lake) processors, 252GB RAM, Nvidia GeForce RTX2080ti (11GB VRAM), and Samsung 860 PRO SSD. For self-influence calculation, we utilize the JAX [3] library for fast Hessian vector product calculation. For all other deep learning functionalities, we utilize Pytorch [42]. In Table 8, the wall-clock duration of each component of our method is shown. We observe that the self-influence calculation step takes a longer time compared to the fine-tuning step due to the intersection process. However, this can be executed in parallel, which reduces the time cost of self-influence calculation approximately threefold. In Table 9, a wall-clock time comparison with the other baselines is shown. Our method consumes a significantly lesser amount of time, dropping to less than half the time of ERM full training when the self-influence calculation is executed in parallel. Reflecting on these results, we assert that the time cost of our method is rather small or even negligible compared to the full training time of other baselines.

Table 8: The average and the standard error of computational costs over three runs. We measure the computing time for full training as the wall-clock time of each component. Self-influence (parallel) represents calculating the bias-conditioned self-influence in GPU-parallel. Note that † indicates that corresponding methods use JAX while others utilize PyTorch.

| Component | Self-influence | Self-influence (parallel) | Fine-tuning |
|---|---|---|---|
| Time (min.) | $11.46^{\dagger}_{\pm0.08}$ | $3.86^{\dagger}_{\pm0.03}$ | $1.08_{\pm0.04}$ |

Table 9: The average and the standard error of computational costs over three runs. We measure the computing time for full training as the wall-clock time of each method. Ours (parallel) presents our method which computes bias-conditioned self-influence in GPU-parallel. Note that † indicates that corresponding methods use JAX while others utilize PyTorch.

| Method | ERM | LfF | DFA | SelecMix | Ours | Ours (parallel) |
|---|---|---|---|---|---|---|
| Time (min.) | $22.55_{\pm0.32}$ | $33.64_{\pm0.34}$ | $53.18_{\pm2.55}$ | $352.53_{\pm5.13}$ | $12.54^{\dagger}_{\pm0.08}$ | $4.94^{\dagger}_{\pm0.03}$ |

## H Analysis of intersections within pivotal sets

In this section, we analyze the effects of intersections between pivotal sets obtained from various random initializations of models. For the comparison, we provide the number of samples, detection precision, and performance after fine-tuning models across different numbers of the intersections

in Table 10, Table 11, and Table 12. We observe that the detection precision increases as the number of intersections rises, while the number of samples in the pivotal set decreases. For the performance, a higher number of intersections shows effectiveness in the highly-biased scenarios, as bias-conflicting samples are scarce, and intersections reduce the size of the pivotal set. In contrast, a fewer intersections exhibit superior performance in low-based scenarios as there are abundant bias-conflicting samples. Note that, to observe the trend across varying ratios of bias-conflicting samples, we conduct experiments on CIFAR10C (0.5%, 1%, 2%, 5%, 20%, 30%, 50%, 70%).

Table 10: The average and the standard error of **the number of pivotal sets** over three runs considering numbers of intersections.

| Number of Intersections | CIFAR10C | | | | | | | |
|---|---|---|---|---|---|---|---|---|
| | 0.5% | 1% | 2% | 5% | 20% | 30% | 50% | 70% |
| 1 | 1000 | 1000 | 1000 | 1000 | 1000 | 1000 | 1000 | 1000 |
| 2 | $322.67_{\pm3.38}$ | $386.67_{\pm11.98}$ | $503.67_{\pm39.75}$ | $577.00_{\pm16.46}$ | $554.00_{\pm63.38}$ | $421.67_{\pm21.17}$ | $309.67_{\pm40.03}$ | $290.00_{\pm70.32}$ |
| 3 | $201.67_{\pm4.91}$ | $267.00_{\pm8.50}$ | $388.33_{\pm19.06}$ | $430.33_{\pm30.66}$ | $452.00_{\pm65.09}$ | $281.00_{\pm11.02}$ | $144.67_{\pm30.99}$ | $141.67_{\pm30.99}$ |

Table 11: The average and the standard error of **detection precision** over three runs considering numbers of intersections.

| Number of Intersections | CIFAR10C | | | | | | | |
|---|---|---|---|---|---|---|---|---|
| | 0.5% | 1% | 2% | 5% | 20% | 30% | 50% | 70% |
| 1 | $13.27_{\pm0.50}$ | $24.90_{\pm0.87}$ | $47.07_{\pm0.65}$ | $76.27_{\pm0.50}$ | $97.10_{\pm0.95}$ | $97.60_{\pm1.20}$ | $91.27_{\pm2.02}$ | $80.83_{\pm1.15}$ |
| 2 | $31.68_{\pm1.61}$ | $52.83_{\pm1.80}$ | $75.17_{\pm3.66}$ | $92.01_{\pm0.80}$ | $99.77_{\pm0.16}$ | $99.02_{\pm0.74}$ | $96.00_{\pm0.47}$ | $83.68_{\pm0.99}$ |
| 3 | $45.57_{\pm1.63}$ | $68.18_{\pm0.96}$ | $86.13_{\pm1.18}$ | $96.60_{\pm0.11}$ | $99.94_{\pm0.06}$ | $99.88_{\pm0.12}$ | $98.30_{\pm0.30}$ | $85.81_{\pm2.05}$ |

Table 12: The average and the standard error of **classification accuracy** of 'Ours+SelecMix' over three runs considering numbers of intersections.

| Number of Intersections | CIFAR10C | | | | | | | |
|---|---|---|---|---|---|---|---|---|
| | 0.5% | 1% | 2% | 5% | 20% | 30% | 50% | 70% |
| 1 | $36.44_{\pm0.34}$ | $40.76_{\pm0.03}$ | $49.57_{\pm0.41}$ | $59.31_{\pm0.15}$ | $\mathbf{67.99}_{\pm0.33}$ | $\mathbf{67.04}_{\pm0.65}$ | $\mathbf{67.39}_{\pm0.79}$ | $\mathbf{70.09}_{\pm0.28}$ |
| 2 | $\mathbf{38.85}_{\pm0.62}$ | $43.47_{\pm0.21}$ | $51.43_{\pm0.53}$ | $\mathbf{60.22}_{\pm0.19}$ | $66.96_{\pm0.25}$ | $65.90_{\pm0.81}$ | $66.77_{\pm0.40}$ | $69.92_{\pm0.53}$ |
| 3 | $38.74_{\pm0.36}$ | $\mathbf{46.18}_{\pm0.33}$ | $\mathbf{52.70}_{\pm0.40}$ | $59.66_{\pm0.31}$ | $66.66_{\pm0.43}$ | $64.51_{\pm1.44}$ | $66.45_{\pm0.28}$ | $69.97_{\pm0.21}$ |

# I   Improving performance in low-bias settings

In CIFAR-10C, as the bias severity decreases from 30% to 90%, the dataset gradually transitions into the low-bias domain, approaching an unbiased state at 90%. This reduction undermines the assumption that the bias is sufficiently malignant, reducing the effectiveness of debiasing methods and allowing ERM to achieve better performance. In this context, to improve the performance of our method when applied to ERM—which leverages a large number of conflicting samples—it is necessary to increase the size of the pivotal set, thereby expanding the number of conflicting samples. As shown in Table 13, expanding the pivotal set can improve performance in low-bias settings. This result implies that we could further enhance performance by adjusting the top-k value if we had access to information regarding bias severity.

Table 13: The average and the standard error of classification accuracy over three runs.

| Method | CIFAR10C | | | |
|---|---|---|---|---|
| | 30% | 50% | 70% | 90% |
| ERM | $65.64_{\pm0.51}$ | $71.33_{\pm0.09}$ | $74.90_{\pm0.25}$ | $76.03_{\pm0.26}$ |
| Ours ERM (k=100) | $65.61_{\pm0.77}$ | $70.61_{\pm0.62}$ | $73.20_{\pm0.35}$ | $73.57_{\pm0.16}$ |
| Ours ERM (k=2000) | $\mathbf{71.25}_{\pm0.34}$ | $\mathbf{74.46}_{\pm0.34}$ | $\mathbf{75.84}_{\pm0.33}$ | $\mathbf{76.14}_{\pm0.23}$ |

## J  Qualitative analysis using Grad-CAM

This section provides qualitative results of our method using Grad-CAM [47] on BFFHQ and Waterbird. For BFFHQ, the target attributes are {young, old} and the bias attributes are {man, woman}. For Waterbird, the target attributes are {waterbird, landbird}, and the bias attributes are {water, land}. In Figure 11 (a) and (c), ERM focuses on biased features such as gender and background. However, ERM combined with our method tends to focus more on task-relevant features including age-related facial features and the birds themselves. This implies that our approach effectively guides the model in prioritizing target attributes over biased ones.

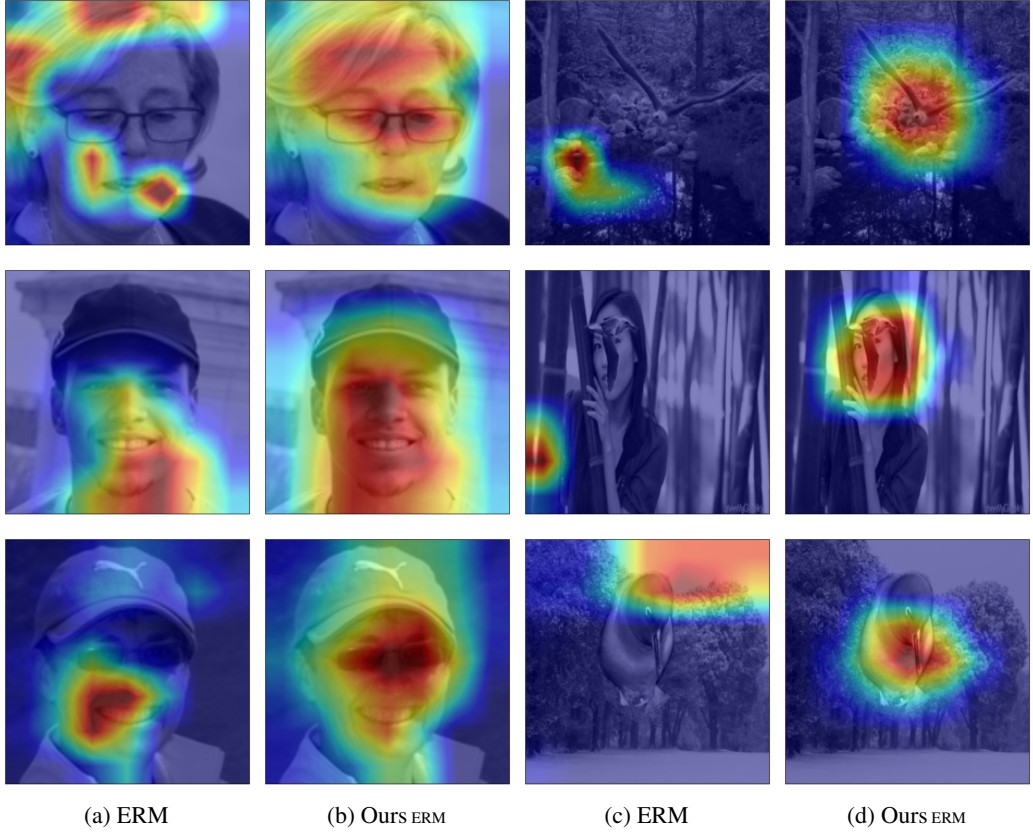

| (a) ERM | (b) Ours ERM | (c) ERM | (d) Ours ERM |

Figure 11: The Grad-CAM of ERM and ERM+Ours on BFFHQ and Waterbird. (a-b) show results on BFFHQ, while (c-d) display results on Waterbird. (a) and (c) represent the Grad-CAMs for ERM, and (b) and (d) correspond to Grad-CAMs for ERM combined with our method.

# K Ablation study on the loss function of the detection model

In this section, we conduct an ablation study on the learning objectives of the detection model. Our method uses Generalized Cross Entropy (GCE), a commonly adopted loss function in debiasing tasks to acquire biased models. However, conceptually, our method can be applied to any loss function to obtain biased models. To demonstrate the generality of our approach with different loss functions, we evaluate it on BFFHQ and Waterbird using alternative objectives, such as GCE, SCE [56], and NCE+RCE [38], which are designed for handling noisy label environments. In Table 14, both SCE and NCE+RCE demonstrate performance comparable to GCE. Since these objectives encourage models to focus more on the majority samples, our method combined with these loss functions also achieves similar results. Note that, naive cross-entropy, which does not promote majority sample utilization, fails on the BFFHQ dataset.

Table 14: The average and the standard error over three runs.

| Method | BFFHQ | Waterbirds |
|---|---|---|
| | 0.5% | 5% |
| SelecMix | $63.07_{\pm 2.32}$ | $74.72_{\pm 1.14}$ |
| Ours w/ CE SelecMix | $62.73_{\pm 3.71}$ | $88.73_{\pm 0.45}$ |
| Ours w/ GCE SelecMix | $65.80_{\pm 3.12}$ | $89.67_{\pm 0.38}$ |
| Ours w/ SCE SelecMix | $66.20_{\pm 0.53}$ | $89.46_{\pm 0.36}$ |
| Ours w/ NCE+RCE SelecMix | $\mathbf{67.73}_{\pm 1.99}$ | $\mathbf{89.72}_{\pm 0.41}$ |

# L Ablation study on Influence estimation methods

We conduct an ablation study on other influence estimation methods. We leverage the fundamental form of Influence Functions to demonstrate the generalizability of our approach. However, other estimation methods are compatible. To further show this, we evaluate our method on BFFHQ and Waterbird using MoSo [50], TracIn [43], and Arnoldi [46]. As shown in Table 15, TracIn outperforms the basic IF, while MoSo and Arnoldi exhibit comparable performance. These results indicate that our method can enhance performance across various estimation approaches.

Table 15: The average and the standard error of detection precision over three runs.

| Method | BFFHQ | Waterbirds |
|---|---|---|
| | 0.5% | 5% |
| SelecMix | $63.07_{\pm 2.32}$ | $74.72_{\pm 1.14}$ |
| Ours SelecMix | $65.80_{\pm 3.12}$ | $89.67_{\pm 0.38}$ |
| Ours w/ MoSo SelecMix | $63.13_{\pm 3.27}$ | $89.72_{\pm 1.12}$ |
| Ours w/ TracIn SelecMix | $\mathbf{69.20}_{\pm 0.50}$ | $\mathbf{90.39}_{\pm 0.70}$ |
| Ours w/ Arnoldi SelecMix | $66.40_{\pm 3.12}$ | $71.08_{\pm 3.12}$ |

# M Evaluation with fairness metrics

To further demonstrate the effectiveness of our method, we evaluate it using fairness metrics including demographic parity (DP) [5] and equalized odds equal opportunity (EOP) [13] on Waterbird. In Table 16, our method significantly improves performance in both DP and EOP. It indicates that our method also addresses the fairness problem.

Table 16: The average and the standard error of demographic parity (DP) and equalized odds equal opportunity (EOP) over three runs.

| Method | Waterbirds | |
|---|---|---|
| | DP ($\downarrow$) | EOP ($\downarrow$) |
| ERM | $0.1826_{\pm 0.0044}$ | $0.2731_{\pm 0.0187}$ |
| SelecMix | $0.1146_{\pm 0.0004}$ | $0.1885_{\pm 0.0100}$ |
| Ours $_{\text{SelecMix}}$ | $\mathbf{0.0242}_{\pm 0.0053}$ | $\mathbf{0.0099}_{\pm 0.0064}$ |

## N  Experimental settings

### N.1  A detailed description of benchmark datasets

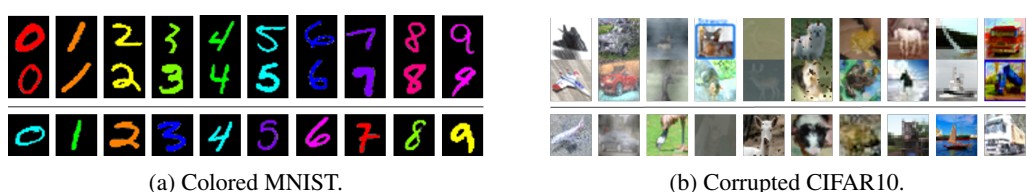

(a) Colored MNIST.  (b) Corrupted CIFAR10.

Figure 12: Example images of CMNIST and CIFAR10C. Images in the first and second rows are *bias-aligned* and images in the third row are *bias-conflicting*.

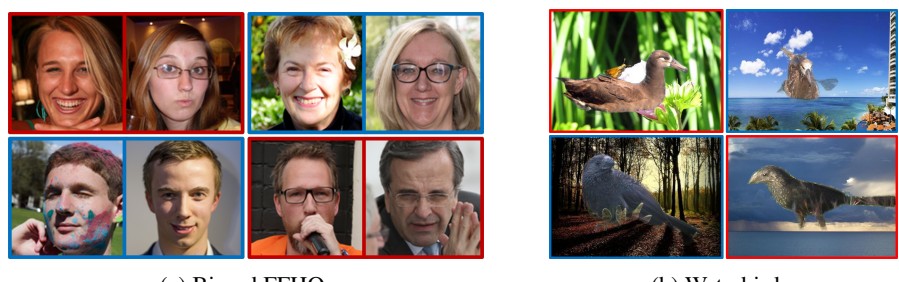

(a) Biased FFHQ.  (b) Waterbirds.

Figure 13: Example images of BFFHQ and Waterbirds. The red-bordered images are *bias-aligned* and the blue-bordered images are *bias-conflicting*.

**Colored MNIST.**  Colored MNIST (CMNIST) is a synthetically modified version of MNIST [6], where the digit is the label and the color is the bias. For example, an image of digit 0 is correlated with the color red. We use the following bias-conflicting ratios: $r \in \{0.5\%, 1\%, 2\%, 5\%\}$. The images are in 28 x 28 resolution and are resized to 32 x 32. There are approximately 55,000 training, 5,000 validation, and 10,000 test samples. Examples are shown in Figure 12(a).

**Corrupted CIFAR10.**  Corrupted CIFAR10 (CIFAR10C) is a synthetically modified version of CIFAR10 [30] proposed by  Hendrycks and Dietterich [17] with the following common corruptions as the bias: {Snow, Frost, Fog, Brightness, Contrast, Spatter, Elastic transform, JPEG, Pixelate and Saturate}. We use the following bias-conflicting ratios: $r \in \{0.5\%, 1\%, 2\%, 5\%, 20\%, 30\%, 50\%, 90\%(\text{unbiased})\}$. The images are in 32 x 32 resolution. There are approximately 45,000 training, 5,000 validation, and 10,000 test samples. Examples are shown in Figure 12(b).

**Biased FFHQ.**  Biased FFHQ (BFFHQ) [31] is a curated Flickr-Faces-HQ (FFHQ) [22] dataset, which consists of images of human faces. The designated task label is the age {young, old} while the bias attribute is the gender {man, woman}. The bias-conflicting ratio is $r \in \{0.5\%\}$. The images

are in 128 x 128 resolution and are resized to 224 x 224. There are approximately 20,000 training, 1,000 validation, and 1,000 test samples. Examples are shown in Figure 13(a).

**Waterbirds.** Waterbirds is proposed by Sagawa et al. [44], which synthetically combines bird images from the Caltech-UCSD Birds-200-2011 (CUB) with place background as bias. It consists of bird images to classify bird types {waterbird, landbird}, but their backgrounds {water, land} are correlated with bird types. The bias-conflicting ratio is $r \in \{5\%\}$. The images are in varying resolutions and are resized to 224 x 224. There are approximately 5,000 training, 1,000 validation, and 6,000 test samples. Examples are shown in Figure 13(b).

**NICO.** NICO is a dataset designed to evaluate non I.I.D. classification by simulating arbitrary distribution shifts. To evaluate debiasing methods, a subset composed of *animal* classes label is utilized, as in [54]. The class labels (*e.g.* "dog") are correlated to spurious contexts (*e.g.* "on grass", "in water", "in cage", "eating", "on beach", "lying", "running") which exhibits a long-tail distribution. The images are in varying resolutions and are resized to 224 x 224. There are approximately 3,000 training, 1,000 validation, and 1,000 test samples.

## N.2 Baselines

We validate our method by combining various debiasing approaches. ERM is the model trained by cross-entropy loss. GroupDRO [44] minimizes the worst-group loss by exploiting group labels directly. LfF [40] detects bias-conflicting samples and allocates large loss weights on them. DFA [31] augments diverse features by swapping the features obtained from the biased model and concatenating the feature from the debiased model with the exchanged feature. BPA [48] utilizes a clustering method to identify pseudo-attributes using a clustering approach and adjusts loss weights according to the cluster size and its loss. DCWP [41] debiases a network by pruning biased neurons. SelecMix [19] identifies and mixes a bias-contradicting pair within the same class while detecting and mixing a bias-aligned pair from different classes. Note that we adopt SelecMix+LfF rather than SelecMix since SelecMix+LfF exhibits superior performance than SelecMix [19].

## N.3 Evaluation protocol

We provide experimental setups for evaluation. We use JAX [3] and PyTorch [42] for the experiments. We conduct our experiments with a machine equipped with Intel Xeon Gold 5215 (Cascade Lake)594 processors, 252GB RAM, Nvidia GeForce RTX2080ti (11GB VRAM) (or Nvidia GeForce RTX3090 (24GB VRAM)), and Samsung 860 PRO SSD. In constructing pivotal sets, we adopt ResNet18 [15] as the base architecture for all datasets. For optimization, we employ the Adam optimizer [25] with a learning rate of 0.001, and train the models for 5 epochs. To calculate self-influence, we only utilize the last layer of the models. In fine-tuning, we deploy ResNet18 for CMNIST, CIFAR10C, BFFHQ, and NICO while ResNet50 is used for Waterbirds as following other baselines [40, 31, 36]. We adopt the Adam optimizer for CMNIST, CIFAR10C, BFFHQ, NICO while SGD is used for Waterbirds. For the learning rate, we use 0.001 for CMNIST, CIFAR10C, Waterbirds, and $10^{-4}$ for BFFHQ. We apply cosine annealing [37] to decay the learning rate to $10^{-3}$ of the initial value. We utilize weight decay of $10^{-4}$ for all datasets. We fine-tune the pre-trained models for 100 iterations. We set $\lambda = 0.1$ for all experiments. For baselines [31, 19], we use the officially released codes. For our method, we adopt $k = 100, \lambda = 0.1$ for all datasets.

## N.4 Licenses for existing assets

Flickr-Faces-HQ (FFHQ) [22] is a high-quality image dataset of human faces, originally created as a benchmark for generative adversarial networks (GAN). The individual images were published in Flickr by their respective authors under either Creative Commons BY 2.0, Creative Commons BY-NC 2.0, Public Domain Mark 1.0, Public Domain CC0 1.0, or U.S. Government Works license. NICO [16] dataset does not own the copyright of images. Only researchers and educators who wish to use the images for non-commercial researches and/or educational purposes, have access to NICO. JAX [3] has Apache License.

