# OpenReview forum: "A Simple Remedy for Dataset Bias via Self-Influence: A Mislabeled Sample Perspective"
_NeurIPS.cc/2024/Conference — NeurIPS 2024 poster_

### Official Review · Reviewer_hyta · 2024-06-25

**Soundness:** 2
**Presentation:** 3
**Contribution:** 2
**Rating:** 6
**Confidence:** 3

**Summary:**

The paper proposes a method for debiasing biased models through fine-tuning on a specific subset with a high portion of unbiased ("bias-conflicting") samples. The paper uses self-influence in the early training epochs to identify this specific subset, where high self-influence is an indicator of a bias-conflicting sample. The intuition behind this approach is that models tend to learn biases early in training, so that samples without the bias feature will have high self-influence scores.

**Strengths:**

- The paper is well-written.
- The hypothesis of fine-tuning on a subset of bias-conflicting samples is grounded on a thorough analysis.

**Weaknesses:**

- There are some unclarities in the method. Please see questions.
- As "bias-aligned" and "bias-conflicting" is new terminology, it is slightly hard to follow as a reader. Adding an example image instead of the description in section 3.2 would help this.
- Formatting:
     - Figure 2: Please add the error bars from your 3 runs.
     - Figure 3: For easier comparability of (b) and (c) please use the same y-axis., same for the histogram’s x-axis of (d).
     - Figure 4: Please use the same x-axis across the subplots. Using your 3 runs, it would be interesting to add error bars to the plot, too.
     - “Figure x” and “Table x” font sizes are larger than the caption font size.
     - Table 1 caption: space between the cross symbol and “denotes” missing
- Minor:
     - Line 33: It’s unclear what “it” refers to.
     - Line 79: Please also include the citation Hampel 1974 that introduced Influence Functions from Robust statistics.

**Questions:**

- Self-influence is often used in mislabeled sample detection as a way to evaluate training data attribution methods. Still, SI mainly indicates that a sample is OOD (which could be a case of mislabeling). Does your approach generalize to OOD cases, then, too?
- What is the difference between IF and SI is? SI has been used in IF and data attribution work commonly as an evaluation protocol to find mislabeled samples.
- Equation 2: It makes sense to detect bias-conflicting samples early in training. Yet, at this stage the model has yet to converge and I can imagine that the estimation quality of the inverse Hessian vector product for self-influence is rather low and uses a large damping factor. What is the damping factor used?
- How does the estimation quality influence your method and results? Would a dynamic training data attribution approach like e.g. TracIn (computing influence as a sum of gradient dot products across training) be an alternative?
- How do you compute self-influence/what kind of HVP estimation algorithm do you use?
- Lines 203-205: How many samples are in the intersection? Is there high variance in the self-influence?
- Line 330 Broader impact: If only biased data is available, how can bias-conflicting samples be found?

**Limitations:**

Limitations and broader impact are discussed in the conclusion.

---

> ### Author Rebuttal · Authors · 2024-08-07
>
> We thank you for your constructive comments.
>
> ---
>
> ### [Q1] Self-influence is often used in mislabeled sample detection as a way to evaluate training data attribution methods. Still, SI mainly indicates that a sample is OOD (which could be a case of mislabeling). Does your approach generalize to OOD cases, then, too?
>
> Yes, our approach generalizes to OOD cases. The core idea of using Self-Influence in mislabeled sample detection is to identify minority samples that contradict the dominant features learned by the model. Since OOD samples, including mislabeled ones, also exhibit a contradiction due to their incorrect labels or the absence of the dominant features learned by the model, our approach is applicable to them as well.
>
> In terms of the degree of contradiction, although we have not thoroughly validated this, it is reasonable to predict the following order: mislabeled samples > non-mislabeled OOD samples > bias-conflicting samples. Since bias-conflicting samples share task-related features with the majority, the degree of contradiction is weaker than that of OOD samples, making their detection a significant challenge.
>
> ---
>
> ### [Q2] What is the difference between IF and SI is?
> In Figures 2 and 3, the notation $IF$ refers to the Influence Function on the training set, as specified in the captions. Please note that IF on the training set is the baseline in our experiments, substituting for IF on the unbiased validation set, which is unavailable in our target problem. To avoid any misunderstanding, we will modify the notation to $IF_{train}$.
>
> To elaborate, typically, if a validation set that resembles the distribution of the test set is available, it is used as $z'$ in Influence Function $I(z, z′)$. However, in challenging situations where an unbiased validation set is not available, Self-Influence $I(z,z)$ is used by setting $z’$ as the training sample $z$ itself instead of the validation set. SI measures how the prediction of $z$ changes when $z$ is excluded from the training set. This means that the higher contradiction the sample $z$ exhibits, the more difficult it becomes for the model to make accurate predictions based on features learned from other samples.
>
> ---
>
> ### [Q3] Equation 2: It makes sense to detect bias-conflicting samples early in training. Yet, at this stage, the model has yet to converge and I can imagine that the estimation quality of the inverse Hessian vector product for self-influence is rather low and uses a large damping factor. What is the damping factor used?
>
> As the reviewer mentioned, since we use an early-stage model, the estimation quality of the inverse Hessian vector product is to some extent insufficient. To compensate for this, we follow a widely used convention by adding the absolute value of the smallest negative eigenvalue plus 0.0001 to avoid negative eigenvalues. Without additional tuning of the damping factor, this setting has empirically shown good performance across various benchmark datasets in our experiments. However, resolving the issue of insufficient estimation quality due to using an early-stage model is crucial for further performance improvements, making it a primary objective in future work.
>
> ---
>
> ### [Q4]  How does the estimation quality influence your method and results? Would a dynamic training data attribution approach like e.g. TracIn (computing influence as a sum of gradient dot products across training) be an alternative?
> We leveraged the fundamental form of Influence Functions to demonstrate the generalizability of our approach. Of course, other forms of Influence function, such as dynamic training data attribution methods like TracIn[1], can also be viable alternatives.
>
> To further demonstrate this, we have conducted additional experiments on BFFHQ and Waterbird using TracIn and MoSo [2], a recent method that leverages gradients from the training process. As shown in the table below, using TracIn results in better performance compared to the basic form of IF, and MoSo demonstrates comparable performance.
>
> |                | BFFHQ           | Waterbird      |
> |----------------|-----------------|----------------|
> | SelecMix       | 63.07 ± 2.32    | 74.72 ± 1.14   |
> | SelecMix + Ours    | 65.80 ± 3.12    | 89.67 ± 0.38   |
> | SelecMix + Ours_MoSo   | 63.13 ± 3.27    | 89.72 ± 1.12   |
> | SelecMix + Ours_TracIn | **69.20 ± 0.50**    | **90.39 ± 0.70**   |
>
> [1] Pruthi et al. "Estimating training data influence by tracing gradient descent." NeurIPS, 2020.
>
> [2] Tan et al. "Data pruning via moving-one-sample-out." NeurIPS, 2023.
>
> ---
>
> ### [Q5] How do you compute self-influence/what kind of HVP estimation algorithm do you use?
> For both efficiency and bias-focused computations, we calculate self-influence using only the last layer (classification layer), with the Hessian inverse computed exactly. By following [3], we limit the Hessian computation to the parameters of the last layer to reduce computational cost. Additionally, considering [4]'s discovery that retraining only the classification layer can achieve debiasing, focusing on the last layer allows for more bias-focused computations.
>
> In the table below, we present experiments with Arnoldi [5], which estimates self-influence utilizing all the parameters in the model. The results show that Arnoldi performs comparably or worse, underscoring the effectiveness of using only the last layer.
>
> |                | BFFHQ           | Waterbird      |
> |----------------|-----------------|----------------|
> | SelecMix       | 63.07 ± 2.32    | 74.72 ± 1.14   |
> | SelecMix + Ours    | 65.80 ± 3.12    | **89.67 ± 0.38**   |
> | SelecMix + Ours_Arnoldi   | **66.40 ± 3.12**    | 71.08 ± 3.12   |
>
> [3] Daxberger et al. "Laplace redux-effortless bayesian deep learning." NeurIPS, 2021.
>
> [4] Kirichenko et al. "Last layer re-training is sufficient for robustness to spurious correlations." arXiv, 2022.
>
> [5] Schioppa et al. "Scaling up influence functions." AAAI, 2022.
>
> ---

---

> ### Author Response · Authors · 2024-08-07
> **Rebuttal by Authors (2/2)**
>
> ### [Q6] Lines 203-205: How many samples are in the intersection? Is there high variance in the self-influence?
> The variance of self-influence is to some extent high due to the usage of the early-stage model. However, the intersection process substantially mitigates this issue and enhances the ratio of bias-conflicting samples in the resulting pivotal set.
>
> We provided an ablation study on the intersection in Appendix G. The study shows that as the ratio of bias-conflicting samples increases up to 20%, the number of samples after the intersection also increases. This indicates that the intersection adaptively adjusts the size of the pivotal set based on the bias-conflicting ratio, as we intended.
>
> ---
>
> ### [Q7] Line 330 Broader impact: If only biased data is available, how can bias-conflicting samples be found?
> We apologize for the confusion. We intended to refer to scenarios where both bias labels and an unbiased validation set are unavailable. We will revise the statement to clarify this point.
>
> ---
>
> ### [Q8] As "bias-aligned" and "bias-conflicting" is new terminology, it is slightly hard to follow as a reader. Adding an example image instead of the description in section 3.2 would help this.
>
> The terms "bias-aligned" and "bias-conflicting" were first proposed by LfF [6] and are widely used in research on spurious correlations. However, to aid readers who may not be familiar with this terminology, as the reviewer suggested, we will provide additional explanations and include example images in Section 3.2.
>
> [6] Nam et al. "Learning from failure: De-biasing classifier from biased classifier." NeurIPS, 2020.
>
> ---
>
> ### [formatting issues and an additional citation]
> As the reviewer suggested, we will revise formatting issues, and include the citation Hampel 1974 in the revision.

---

> > ### Comment · Reviewer_hyta · 2024-08-09
> >
> > Thank you for your answers to the many questions I posed. I am optimistic about this work and find the research direction of finding bias-conflicting samples important as this is probably a widespread contributor to spurious correlations in learned models. However, some concerns remain: I agree with reviewers BCqo and yem8 that the current evaluation and experiments presented do not test for debiasing per se. I believe that expanding on the Grad-Cam experiments for qualitative analysis and adding a quantitative analysis with e.g. error-parity experiments or spurious Imagenet/waterbirds will improve the paper for a future version. Due to these concerns, I will keep the current score of 4.

---

> ### Author Response · Authors · 2024-08-09
>
> We sincerely appreciate the reviewer’s time and effort in carefully considering our response.
>
> ---
>
> The reviewer expressed concerns that the current quantitative evaluation and experiments do not test for debiasing per se. **However, the quantitative evaluation approach we have presented aligns with the primary and standard evaluation convention in the spurious correlation (debiasing) domain, which assesses the degree of debiasing using an unbiased test set or by measuring average group accuracy and worst-group accuracy [1-11].** Evaluating on an unbiased test set effectively tests for debiasing per se, as it determines whether the model's predictions are driven by genuine task-related features rather than a specific malignant bias. For example, if a model is biased towards a particular attribute, it will likely perform well only on subgroups with that attribute, resulting in lower accuracy on the unbiased test set. **As mentioned in the evaluation protocols in Sec 5.1, we also adhered to the convention by using worst-group accuracy in Waterbird and bias-conflicting accuracy in BFFHQ as the evaluation metrics, further reinforcing the test for debiasing per se.**
>
> To the best of our knowledge, there is currently no qualitative evaluation method in the spurious correlation community that is considered more reliable than these quantitative evaluations. While methods like Grad-CAM to visualize activation map changes or t-SNE to examine how class instances cluster in latent space post-debiasing are sometimes used, these techniques are typically employed as supplements to quantitative results and are not considered more accurate than the quantitative evaluations that we, along with most previous works, have presented using an unbiased test set.
>
> Additionally, **we conducted experiments on five benchmark datasets and further tested on CIFAR-10C by adjusting the severity of bias to levels closer to unbiased conditions.** **This extensive evaluation not only covers a broader range of datasets and scenarios but also provides a more rigorous and comprehensive validation of our method compared to other existing works.**
>
> We once again sincerely thank the reviewer for the thoughtful feedback and the time dedicated to reviewing our work. We hope our responses have addressed the additional concern raised and welcome any further questions.
>
> [1] Nam et al. "Learning from failure: De-biasing classifier from biased classifier." NeurIPS, 2020.
>
> [2] Sagawa et al. "Distributionally robust neural networks.", ICLR, 2020.
>
> [3] Lee et al. "Learning debiased representation via disentangled feature augmentation." NeurIPS, 2021.
>
> [4] Liu et al. "Just Train Twice: Improving Group Robustness without Training Group Information." ICML, 2021.
>
> [5] Seo et al. "Unsupervised learning of debiased representations with pseudo-attributes." CVPR, 2022.
>
> [6] Hwang et al. "Selecmix: Debiased learning by contradicting-pair sampling." NeurIPS, 2022.
>
> [7] Park et al. "Training debiased subnetworks with contrastive weight pruning." CVPR, 2023.
>
> [8] Lim et al., "Biasadv: Bias-adversarial augmentation for model debiasing." CVPR, 2023.
>
> [9] Deng et al. "Robust Learning with Progressive Data Expansion Against Spurious Correlation." NeurIPS, 2023.
>
> [10] Ahn et al. "Mitigating dataset bias by using per-sample gradient." ICLR, 2023.
>
> [11] Jung et al. "Fighting fire with fire: Contrastive debiasing without bias-free data via generative bias-transformation", ICML, 2023.

---

> > ### Comment · Reviewer_hyta · 2024-08-13
> >
> > I apologize for the late reply and thank the authors for their explanations and additional experiments posted during the discussion period. I think that they show the promise of this direction, and I raise my score accordingly from 4->6.

---

> > > ### Author Response · Authors · 2024-08-13
> > >
> > > Thank you for taking the time to review our paper and provide valuable feedback. We are grateful for your decision to raise the score to 6.

---

### Official Review · Reviewer_yem8 · 2024-07-06

**Soundness:** 2
**Presentation:** 2
**Contribution:** 2
**Rating:** 5
**Confidence:** 5

**Summary:**

This paper tackles the problem of learning generalized models from biased data by detecting mislabeled samples. The authors use BCSI (SI estimated on a trained model with GCE) to detect bias-conflicting samples and construct a pivotal subset based on the BCSI scores of these samples for correcting the biased model without access to a clean validation set.

**Strengths:**

- Originality and Significance: Rather than computing influence scores on a clean validation set (which is usually not feasible to obtain in practice), this paper uses self-influence estimated by the model trained with GCE to detect bias-conflicting samples.
- Quality and Clarity: Since the proposed method heavily relies on the assumption that if the loss increases when a sample is removed from the training set, that sample is likely to be mislabeled, the empirical analysis of self-influence (SI) in detecting bias-conflicting samples is sufficient across four different tasks. The experiments, including ablation studies, are comprehensive.

**Weaknesses:**

- The concept of bias-conflicting samples is not equivalent to unbiased samples, so in the introduction, *“by first identifying bias-conflicting (unbiased) samples”* should be rephrased.
- Some grammar errors:
  - Section 3: “A analysis” => “An analysis”.
  - “trainset” => “training set”.
- The bias issue introduced in this paper is more about robustness rather than fairness since no sensitive attributes (such as gender or race) are included in the problem formulation, and no fairness evaluation (demographic parity or equalized odds) is conducted in the experiment. I recommend the authors change the terminology from fairness to robustness for a more precise expression.
- In Section 3.2, I strongly recommend introducing the concepts of “mislabeled samples,” “bias-conflicting samples,” and “bias-aligned samples” in Section 2.1 for better readability.
- In *“Note that since an unbiased validation set is unavailable in our target problem, we additionally estimate the influence score on the training set, indicated as IF in Figure 2.”* The measure of IF is problematic since it should be obtained via a clean validation set, which is not equivalent to the training set. The authors could conduct the experiment in a noisy label setting, for example, treating a subset as a clean validation set and flipping the others with a noise rate.
- *“GCE emphasizes samples that are easier to learn, amplifying the model’s bias by giving more weight to bias-aligned samples in the training set.”* GCE does prioritize samples that are easier to learn, but this does not necessarily mean that these samples are all bias-aligned.
- The proposed method BCSI in Equation 2 is just SI, it appears to have limited novelty.

**Questions:**

From my understanding, the pivotal subset is used to correct and mitigate bias in the model. So why do the authors mention “use pivotal set (bias-conflicting samples) to recover a biased model”? What the paper actually does is use bias-conflicting samples to correct the biased model. This is very confusing. Is this a typo? Or should it be rephrased to “recover an unbiased model from a biased one” for better clarity?

**Limitations:**

The authors have adequately addressed the limitations. Since this work is mainly about dataset bias, so I do not see any negative societal impact.

---

> ### Author Rebuttal · Authors · 2024-08-07
>
> We appreciate your valuable feedback.
>
> ---
>
> ### [Q1] The bias issue introduced in this paper is more about robustness rather than fairness since no sensitive attributes (such as gender or race) are included in the problem formulation, and no fairness evaluation (demographic parity or equalized odds) is conducted in the experiment. I recommend the authors change the terminology from fairness to robustness for a more precise expression.
>
> As the reviewer mentioned, the term robustness is more precise since our target problem is addressing malignant biases (spurious correlations) in the dataset that prevent the model from learning task-related features.
>
> However, there is also an intersection with fairness, especially when considering cases where the malignant bias involves sensitive attributes such as gender or race. Numerous related studies addressing spurious correlations [1,2,3,4,5,6] have shown that mitigating these correlations aids in achieving fairness. For example, in the Biased FFHQ (BFFHQ) dataset, which is one of the benchmark datasets consisting of images of human faces, the designated task label is age {young, old} while the bias attribute is gender {man, woman}. In addition, we conduct experiments on CelebA, a more widely used benchmark dataset in the fairness domain. As shown in the table below, our method also demonstrated effective performance on this dataset.
>
> |   CelebA             |  Averaged Acc. | Worst Group Acc. |
> |----------------|-----------------|----------------|
> | ERM| 95.32 ± 0.34    | 45.19 ± 0.67   |
> | JTT | 90.14 ± 0.61    | 72.22 ± 2.51   |
> | JTT + Ours   | 86.19 ± 1.32    | **80.17 ± 1.29**   |
>
>
> [1] Nam et al. "Learning from failure: De-biasing classifier from biased classifier." NeurIPS, 2020.
>
> [2] Lee et al. "Learning debiased representation via disentangled feature augmentation." NeurIPS, 2021.
>
> [3] Seo et al. "Unsupervised learning of debiased representations with pseudo-attributes." CVPR, 2022.
>
> [4] Hwang et al. "Selecmix: Debiased learning by contradicting-pair sampling." NeurIPS, 2022.
>
> [5] Park et al. "Training debiased subnetworks with contrastive weight pruning." CVPR, 2023.
>
> [6] Deng et al. "Robust Learning with Progressive Data Expansion Against Spurious Correlation", NeurIPS, 2023.
>
> ---
>
> ### [Q2] In Section 3.2, I strongly recommend introducing the concepts of “mislabeled samples,” “bias-conflicting samples,” and “bias-aligned samples” in Section 2.1 for better readability.
>
> To aid readers who may not be familiar with this terminology, as the reviewer suggested, we will introduce these concepts in Sec 2.1 for better readability.
>
> ---
>
> ### [Q3] In “Note that since an unbiased validation set is unavailable in our target problem, we additionally estimate the influence score on the training set, indicated as IF in Figure 2.” The measure of IF is problematic since it should be obtained via a clean validation set, which is not equivalent to the training set. The authors could conduct the experiment in a noisy label setting, for example, treating a subset as a clean validation set and flipping the others with a noise rate.
>
> In Figures 2 and 3, the notation $IF$ refers to the Influence Function on the training set, as specified in the captions. Please note that IF on the training set is the naive baseline in our experiments, substituting for IF on the unbiased validation set. While IF on the unbiased validation set would be a stronger baseline, it is not available in our target problem. To avoid any misunderstanding, we will modify the notation to $IF_{train}$.
>
> ---
>
> ### [Q4] “GCE emphasizes samples that are easier to learn, amplifying the model’s bias by giving more weight to bias-aligned samples in the training set.” GCE does prioritize samples that are easier to learn, but this does not necessarily mean that these samples are all bias-aligned.
>
> We apologize for any confusion. We agree that GCE does not necessarily imply that all these samples are bias-aligned, and our intention was to highlight this tendency. We will revise the statement to: "GCE emphasizes samples that are easier to learn, thereby amplifying the model’s bias by tending to give more weight to bias-aligned samples in the training set.”
>
> ---
>
> ### [Q5] So why do the authors mention “use pivotal set (bias-conflicting samples) to recover a biased model”?
>
> We apologize for the confusion. Our intention was to convey the meaning of "rectify a biased model." We will revise it.
>
> ---
>
> ### [Minor issues]
>
> As the reviewer suggested, we will revise grammar errors, and rephrase “by first identifying bias-conflicting (unbiased) samples”  in the revision.

---

> > ### Author Response · Authors · 2024-08-12
> >
> > To further address the reviewer's concerns about fairness, we conduct experiments on the text dataset MultiNLI (experiments on the CivilComments dataset are currently running due to time constraints), which is an widely used benchmark dataset in the fairness domain. Specifically, MultiNLI involves classifying the relationship between two sentences, with the bias label being the presence of negation words in the second sentence, often linked to the contradiction label.
> >
> > As shown in the table below, our method effectively works on a text dataset as well. We will include the complete table in the revision.
> >
> > | MultiNLI | Avg Acc. | Worst-group Acc. |
> > |---|---|---|
> > | JTT | 80.0 | 70.2 |
> > | JTT+Ours | 79.8 | **73.6** |

---

> > > ### Comment · Reviewer_yem8 · 2024-08-13
> > >
> > > Thanks for the author's detailed response to each of my questions and concerns. The use of the self-influence function to identify bias-conflicting samples is well-motivated. I think the fairness evaluation for this work is more align with Rawlsian Max-Min fairness, and I really appreciate the supplementary experiments that demonstrate this. However, I still believe this paper needs further improvements regarding the clarity of definitions, explicit addressing of fairness metrics, and the novelty of the work, so I keep the current rating score.

---

> ### Author Response · Authors · 2024-08-13
>
> Thank you for your comments, and we would like to provide a response to the reviewer's remaining concerns.
>
> ---
>
> ### [The clarity of definitions] ###
>
> **We carefully defined the terms in Sec. 3 using formulas following the conventions.** However, if any definitions remain unclear, we would be more than willing to clarify them further (e.g., by including additional images for bias-aligned and conflicting samples [3]) to improve readability. We would be grateful if you could specify which definitions are unclear, as your feedback will help us to further enhance the clarity and quality of our paper.
>
> ---
>
> ### [Explicit addressing of fairness metrics] ###
>
> The primary goal of our paper is to address spurious correlations in training datasets. Therefore, our primary evaluation focus is on assessing how well the trained model predicts on test datasets that are free from the spurious correlations present in the training set, using task-related features.
>
> **We have employed the standard and widely accepted benchmark datasets and evaluation metrics in spurious correlation studies [1-20]. It is important to note that the vast majority of spurious correlation studies[1-20] used evaluation methods the same as ours, such as unbiased accuracy and worst(or minority)-group accuracy, rather than the fairness metrics commonly used in the fairness domain.**
>
> However, we agree that incorporating fairness metrics would further enhance the evaluation section. Accordingly, we share experimental results evaluated using demographic parity (DP) and ~~equalized odds~~  equal opportunity (EOP) metrics on the Waterbird dataset. As shown in the table below, our approach demonstrates clear performance improvements, even when evaluated using fairness metrics.
>
> | Waterbird | DP | EOP |
> |:---:|:---:|:---:|
> | ERM | 0.1826 ± 0.0044 | 0.2731 ± 0.0187 |
> | SelecMix | 0.1146 ± 0.0004 | 0.1885 ± 0.0100 |
> | SelecMix+Ours | **0.0242 ± 0.0053** | **0.0099 ± 0.0064** |
>
> ---
>
> We appreciate the reviewer’s valuable feedback and the time spent reviewing our work. We hope that our responses have addressed the remaining concerns raised.
>
> ---
>
> [1] Wang et al. "Learning robust representations by projecting superficial statistics out." ICLR, 2019.
>
> [2] Bahng et al. "Learning de-biased representations with biased representations." ICML, 2020.
>
> [3] Nam et al. "Learning from failure: De-biasing classifier from biased classifier." NeurIPS, 2020.
>
> [4] Sagawa et al. "Distributionally robust neural networks.", ICLR, 2020.
>
> [5] Liu et al. "Just Train Twice: Improving Group Robustness without Training Group Information." ICML, 2021.
>
> [6] Kim et al. "Biaswap: Removing dataset bias with bias-tailored swapping augmentation." ICCV, 2021.
>
> [7] Lee et al. "Learning debiased representation via disentangled feature augmentation." NeurIPS, 2021.
>
> [8] Hong et al. "Unbiased classification through bias-contrastive and bias-balanced learning." NeurIPS, 2021.
>
> [9] Nam et al. "Spread spurious attribute: Improving worst-group accuracy with spurious attribute estimation."  ICLR, 2022.
>
> [10] Seo et al. "Unsupervised learning of debiased representations with pseudo-attributes." CVPR, 2022.
>
> [11] Idrissi et al. "Simple data balancing achieves competitive worst-group-accuracy." CLeaR, 2022.
>
> [12] Hwang et al. "Selecmix: Debiased learning by contradicting-pair sampling." NeurIPS, 2022.
>
> [13] Kim et al. "Learning debiased classifier with biased committee." NeurIPS, 2022.
>
> [14] Park et al. "Training debiased subnetworks with contrastive weight pruning." CVPR, 2023.
>
> [15] Lim et al., "Biasadv: Bias-adversarial augmentation for model debiasing." CVPR, 2023.
>
> [16] Deng et al. "Robust Learning with Progressive Data Expansion Against Spurious Correlation." NeurIPS, 2023.
>
> [17] Ahn et al. "Mitigating dataset bias by using per-sample gradient." ICLR, 2023.
>
> [18] Kirichenko et al.. "Last layer re-training is sufficient for robustness to spurious correlations." ICLR, 2023.
>
> [19] Liu et al. "Avoiding spurious correlations via logit correction." ICLR, 2023.
>
> [20] Jung et al. "Fighting fire with fire: Contrastive debiasing without bias-free data via generative bias-transformation", ICML, 2023.
>
> ---

---

> > ### Comment · Reviewer_yem8 · 2024-08-13
> >
> > I greatly appreciate the authors’ response in providing supplementary experimental results and explanations to address my remaining concerns with such a short time. As I mentioned, I really like the idea of this paper, so if the authors can guarantee improvements to the manuscript, I will raise my score.

---

> > > ### Author Response · Authors · 2024-08-13
> > >
> > > We sincerely appreciate the reviewer for the time and effort dedicated to reviewing our paper. We guarantee that all the discussions and improvements made during the rebuttal period will be incorporated into the manuscript. We are deeply grateful for the valuable feedback, which has significantly strengthened our paper.

---

> > > > ### Comment · Reviewer_yem8 · 2024-08-13
> > > >
> > > > Thanks for the authors' response, and I have raised my score.

---

> > > > > ### Author Response · Authors · 2024-08-13
> > > > >
> > > > > We are truly grateful for the decision to increase the score.
> > > > > We apologize for any confusion and confirm that EOP stands for Equal Opportunity, not Equalized Odds. We will ensure this clarification is clearly addressed in the revision.

---

> ### Comment · Reviewer_yem8 · 2024-08-13
>
> By the way, I think Reviewer BCqo meant Equal Opportunity (EOP), not Equalized Odds (EO). These two metrics are slightly different. Please ensure the authors are using the correct metric for evaluation.

---

### Official Review · Reviewer_BCqo · 2024-07-13

**Soundness:** 3
**Presentation:** 2
**Contribution:** 3
**Rating:** 5
**Confidence:** 4

**Summary:**

The authors focus on detecting bias-conflicting samples to recover biased models. They propose a Bias-Conditioned Self-Influence to help identify bias-conflicting samples in the early stage of model training. Experiments on public datasets are conducted to demonstrate the effectiveness of the proposed method.

**Strengths:**

1. The introduced perspective of using a bias-conditioned self-influence for bias-conflicting sample detection is interesting;

2. The experimental results look promising.

**Weaknesses:**

**Majors:**

1. The paper aims to rectify bias within a model. However, only the accuracy of models and distribution of BCSI scores are provided. Additional experiments are needed to demonstrate that the bias within a model could be reduced by the proposed method.

2. The authors employ Generalized Cross Entropy to get a more biased model. What about the performance with other losses?

3. Why does ERM perform better than the proposed method in Table 2? An insightful analysis is needed.

4. I suggest the authors to reorganize the paper to make it easier to follow. See minors for details.


**Minors:**

1. Table 2 is mentioned before Table 1;

2. Figure 2 mentioned before Figure 1;

3. Experimental settings are mentioned in the technical part ($\lambda=0.1$).

5. There are some typos. For example, on Page 3, Line 102: A analysis of ...

**Questions:**

Please see the Weaknesses part and kindly correct me if there are any misunderstandings.

---

> ### Author Rebuttal · Authors · 2024-08-07
>
> We thank you for the constructive comments.
>
> ---
>
> ### [Q1] The paper aims to rectify bias within a model. However, only the accuracy of models and distribution of BCSI scores are provided. Additional experiments are needed to demonstrate that the bias within a model could be reduced by the proposed method.
>
> To further demonstrate the effectiveness of our framework in reducing model bias, we have conducted Grad-CAM [1] analysis on the BFFHQ and Waterbird datasets. In the BFFHQ dataset, the target attribute set is {young, old} and the bias attribute set is {man, woman}. For the Waterbird dataset, the target attribute set is {waterbird, landbird} and the bias attribute set is {water, land}.
>
> As shown in Figure 1 of the attached PDF in the global response, the biased models (a) and (c) tend to focus on biased attributes such as gender and background. However, when applying our method, as illustrated in (b) and (d), the model's attention shifts to more task-related features, such as age in faces and bird species. This indicates that our method effectively redirects the model’s focus away from biased attributes and toward the target attributes.
>
> [1] Selvaraju et al. "Grad-cam: Visual explanations from deep networks via gradient-based localization." CVPR, 2017.
>
> ---
>
> ### [Q2] The authors employ Generalized Cross Entropy to get a more biased model. What about the performance with other losses?
>
> We utilize Generalized Cross Entropy (GCE) since it is commonly used in the debiasing domain to obtain a biased model. However, other loss functions can also be viable alternatives. To demonstrate this, we conducted additional experiments on the BFFHQ and Waterbird datasets, employing other loss functions that are follow-ups to GCE, such as SCE [2] and NCE+RCE [3], which are designed for noisy label settings.
>
> As shown in the table below, we present the performance of our method with each applied loss function. Both SCE and NCE+RCE exhibit performance comparable to or slightly better than GCE in our method. These loss functions encourage the model to focus more on the majority of normal samples rather than the minority noisy ones, which also results in a more biased model in the given bias setting.
>
> |              |  BFFHQ | Waterbird |
> |----------------|-----------------|----------------|
> | SelecMix | 63.07± 2.32    | 74.72 ± 1.14   |
> | SelecMix + Ours_CE |  62.73 ± 3.71 | 88.73 ± 0.45 |
> | SelecMix + Ours_GCE | 65.80 ± 3.12 | 89.67 ± 0.38 |
> | SelecMix + Ours_SCE | 66.20 ± 0.53 | 89.46 ± 0.36 |
> | SelecMix + Ours_NCE+RCE |  **67.73 ± 1.99** | **89.72 ± 0.41**  |
>
>
> [2] Wang et al. "Symmetric cross entropy for robust learning with noisy labels." ICCV, 2019.
>
> [3] Ma et al. "Normalized loss functions for deep learning with noisy labels." ICML, 2020.
>
> ---
>
> ### [Q3] Why does ERM perform better than the proposed method in Table 2? An insightful analysis is needed.
>
> Recent debiasing methods are typically designed under the assumption that the bias is malignant enough to mislead a model into extensively relying on the bias to produce a biased predictor. Consequently, in the cases of 70% and 90% in Table 2, the dataset is nearly an unbiased set, breaking this assumption. This leads to the opposite effect, where important samples for learning are disregarded. Ensuring robust performance even when given such unbiased datasets remains an important future goal for the debiasing community.
>
> ---
>
> ### [Minor issues]
> As the reviewer suggested, we will revise the organization of the paper and correct any typos to make it easier to follow.

---

> > ### Comment · Reviewer_BCqo · 2024-08-12
> >
> > Thanks for conducting additional experiments and providing the response. However, I still have concerns as follows:
> >
> > **For Q1**: What about fairness metrics (DP, EOP)?
> >
> > **For Q3**: The response only discussed the cases of 70% and 90%. However, in Table 2, ERM performs better than the proposed method in most settings (30%, 50%, 70%, and 90%).

---

> ### Author Response · Authors · 2024-08-13
>
> We thank you for the detailed review of our work and for your help in improving it further.
>
> ---
>
> ### [Q1.] What about fairness metrics (DP, EOP)? ###
>
> As the reviewer recommended, we additionally evaluate our method on Waterbird using fairness metrics such as DP and EOP. Note that we evaluate ours solely on Waterbird since CMNIST, CIFAR10-C, and NICO have more than two classes, and that BFFHQ's test set contains only bias-conflicting samples. As shown in the table below, our approach demonstrates clear performance improvements, even when evaluated using fairness metrics.
>
> | Waterbird | DP | EOP |
> |:---:|:---:|:---:|
> | ERM | 0.1826 ± 0.0044 | 0.2731 ± 0.0187 |
> | SelecMix | 0.1146 ± 0.0004 | 0.1885 ± 0.0100 |
> | SelecMix+Ours | **0.0242 ± 0.0053** | **0.0099 ± 0.0064** |
>
> ---
>
> ### [Q3.]   The response only discussed the cases of 70% and 90%. However, in Table 2, ERM performs better than the proposed method in most settings (30%, 50%, 70%, and 90%). ###
>
> We apologize for the insufficient response.
>
> In the case of CIFAR-10C, as the bias severity decreases from 30% to 90%, the dataset gradually transitions into the low-bias domain, ultimately approaching an unbiased state at 90%. As mentioned in our previous response, this reduction in bias severity undermines the assumption that the bias is sufficiently malignant, resulting in reduced effectiveness of previous debiasing methods and allowing ERM to achieve better performance.
>
> In this context, to improve the performance of our method when applied to ERM—which leverages a large number of conflicting samples—it is necessary to increase the size of the pivotal set, thereby expanding the number of conflicting samples that our method can utilize.
>
> As demonstrated in the table below, expanding the pivotal set can lead to performance improvements even in low-bias settings, achieving state-of-the-art (SOTA) performance. Furthermore, if we had access to information regarding bias severity (i.e., the proportion of bias-conflicting samples), we could further optimize performance by adjusting the top-k value.
>
> |  | CIFAR10C-30% | CIFAR10C-50% | CIFAR10C-70% | CIFAR10C-90% |
> |---|---|---|---|---|
> | ERM | 65.64 ± 0.51 | 71.33 ± 0.09 | 74.90 ± 0.25 | 76.03 ± 0.26 |
> | ERM+Ours (topk=100) | 65.61 ± 0.77 | 70.61 ± 0.62 | 73.20 ± 0.35 | 73.57 ± 0.16 |
> | ERM+Ours (topk=2000) | **71.25 ± 0.34** | **74.46 ± 0.34** | **75.84 ± 0.33** | **76.14 ± 0.23** |
>
> We appreciate the reviewer for highlighting this point, which has contributed to enhancing the rigor of our paper. We will include a discussion of these findings in the revision.
>
> ---

---

> > ### Comment · Reviewer_BCqo · 2024-08-13
> >
> > I appreciate the authors' efforts in providing a detailed response addressing most of my concerns. I will raise my rating from 4 to 5.

---

> > > ### Author Response · Authors · 2024-08-13
> > >
> > > We sincerely appreciate the reviewer’s insightful suggestions and the decision to raise the score.

---

### Official Review · Reviewer_xCjP · 2024-07-23

**Soundness:** 2
**Presentation:** 3
**Contribution:** 3
**Rating:** 5
**Confidence:** 3

**Summary:**

The authors propose a method to tackle spurious correlations by using influence functions. Specifically, they compute the self-influence on the training set -- the amount that a particular sample's loss changes when it is removed from the training set. Samples with the highest self-influence are then assumed to be in the bias-conflicting (minority group). Then, models can be finetuned, up-weighting this set of identified samples, to obtain a debiased model. The authors benchmark their method on typical spurious correlation benchmarks, finding that they outperform the baselines.

**Strengths:**

- The paper is well-written and easy to follow.
- The proposed method is intuitive, and
- The method outperforms the baselines on typical benchmark datasets.

**Weaknesses:**

1. The proposed method has no theoretical justifications, and so it is unclear under what circumstances it would fail.

2. The authors should evaluate on a few more datasets from the spurious correlation domain, such as CelebA, MultiNLI, and CivilComments. It would be particularly important to demonstrate that the method can work on text datasets. The authors should also add JTT [1] as a baseline.

3. If compute allows, the authors should compute the ablations (Figure 6) for all other datasets. In addition, it would be interesting to show particular samples in the top-k set. Do the samples differ visually as their ranking decreases? How do the samples which are in the top-k set but are not bias-conflicting look?

4. Once the candidate set of bias conflicting samples is identified, there are many other approaches that could be taken. The authors use a simple upweighting approach, but one could e.g. apply GroupDRO to the dataset, with two groups, as well. The authors should benchmark a few of these alternatives.

5. In Table 1, the proposed method only outperforms the baselines when it is used to finetune a model which has had another debiasing approach applied to it, i.e. Ours_ERM underperforms the baselines. Thus, the method requires a decent starting point to work well.

6. The authors should discuss some of the failure modes of the method. One potential failure mode seems to be when the dataset actually contains mislabeled samples, and so the identified set consists of mislabeled samples instead of bias-conflicting samples. The authors should benchmark their method in these settings, potentially with synthetic noise.

[1] https://arxiv.org/pdf/2107.09044

**Questions:**

1. Was there any model selection required in the experiments? If so, how was it done?

2. Why were some baselines omitted from Tables 2 and 3?

**Limitations:**

The authors have adequately addressed the limitations.

---

> ### Author Rebuttal · Authors · 2024-08-07
>
> We sincerely appreciate your time and effort to review our paper.
>
> ---
>
> ### [Q1] The proposed method has no theoretical justifications, and so it is unclear under what circumstances it would fail.
> Although we did not provide theoretical justification, we demonstrated the effectiveness of our method across various settings and datasets. For the circumstances where our method would fail, our method might fail when there is a significant number of mislabeled samples. Specifically, the intuition behind our method is that generic features for bias-conflicting samples are learned later in the training process, and by using the model in its early stage, we can identify these samples through self-influence. If mislabeled samples are present in the dataset, these samples would also exhibit high self-influence, making bias-conflicting samples difficult to distinguish.
>
> ---
>
> ### [Q2] The authors should evaluate on a few more datasets from the spurious correlation domain. The authors should also add JTT as a baseline.
> We conduct an additional experiment on CelebA, including JTT as a baseline. As shown in the table below, our method significantly improves the performance of JTT, exhibiting the effectiveness of our method on datasets with spurious correlations. We are currently running experiments on the MultiNLI and CivilComments and will report the results during the remaining discussion period.
> |CelebA|Avg Acc.|Worst-group Acc.|
> |---|---|---|
> |ERM|95.32$\pm$0.34|45.19$\pm$0.67|
> |JTT|90.14$\pm$0.61|72.22$\pm$2.51|
> |JTT+Ours|86.19$\pm$1.32|**80.17$\pm$1.29**|
>
> ---
>
> ### [Q3] If compute allows, the authors should compute the ablations (Figure 6) for all other datasets.
>
> For the ablation study, we will report the results for other datasets during the remaining discussion period.
>
> ---
>
> ### [Q4] In addition, it would be interesting to show particular samples in the top-k set. Do the samples differ visually as their ranking decreases? How do the samples that are in the top-k set but are not bias-conflicting look?
>
> In Figure 2 of the global response, we provide examples of bias-conflicting and bias-aligned samples from BFFHQ, based on their BCSI scores. All images belong to the top 100 samples according to BCSI. Specifically, (a) shows bias-conflicting samples with high BCSI. (b) indicates bias-conflicting samples with low BCSI. (c) denotes bias-aligned samples with high BCSI. Note that, in BFFHQ, the target attribute set is {young, old} and the bias attribute is {man, woman}. There is a spurious correlation as young women and old men. First, we will analyze only the first row of Figure 2. In (a), the bias-conflicting sample with high BCSI depicts old women, who appear younger compared to the woman in (b). Additionally, bias-aligned samples with high BCSI in (c), depict older men who appear more like women due to makeup, thus having higher BCSI.
>
> ---
>
> ### [Q5] Once the candidate set of bias-conflicting samples is identified, there are many other approaches that could be taken. The authors use a simple upweighting approach, but one could e.g. apply GroupDRO to the dataset, with two groups, as well.
>
> Thank you for your constructive suggestion. To demonstrate the generalizability of our method, we leverage the most fundamental approach of upweighting. As the reviewer mentioned, various methods to leverage the candidate set are certainly applicable. (However, since bias labels are still not provided, employing other methods like GroupDRO would require additional modules.)
>
> ---
>
> ### [Q6] In Table 1, the proposed method only outperforms the baselines when it is used to finetune a model which has had another debiasing approach applied to it, i.e. Ours_ERM underperforms the baselines. Thus, the method requires a decent starting point to work well.
>
> As we consistently fine-tune models for only a few iterations across all models due to the unknown severity of bias, this approach may be insufficient for heavily biased models like ERM. We have empirically observed that additional iterations lead to further performance improvements in such cases. Furthermore, a key advantage of our method is that it is complementary to existing methods and can further rectify models that have already undergone recent debiasing techniques.
>
> ---
>
> ### [Q7] The authors should discuss some of the failure modes of the method. One potential failure mode seems to be when the dataset actually contains mislabeled samples, and so the identified set consists of mislabeled samples instead of bias-conflicting samples.
> As the reviewer mentioned, when the dataset contains mislabeled samples, our method might fail to detect bias-conflicting samples because both bias-conflicting samples and mislabeled samples possess high self-influence values. We will discuss this in our revision. However, this issue is not unique to our method; many recent debiasing methods also fail in the presence of mislabeled samples, as they amplify the learning signal of samples based on loss or gradient values, which can mislead models under these conditions.
>
> ---
>
> ### [Q8] Was there any model selection required in the experiments? If so, how was it done?
> We used the same hyperparameters across all datasets and settings in the main experiments and consistently selected the model from the last epoch (The hyperparameters are provided in Appendix H.3). If an unbiased validation set were available, hyperparameters could be tuned, potentially leading to further performance improvements.
>
> ---
>
> ### [Q9] Why were some baselines omitted from Tables 2 and 3?
> These baselines were omitted from Tables 2 and 3 because they did not conduct experiments on these benchmark datasets, requiring us to perform the hyperparameter search to obtain reasonable performance. Additionally, these methods exhibit a significant performance gap compared to the state-of-the-art method. To further demonstrate the effectiveness of our method, we will report the results in the remaining discussion phase.

---

> ### Author Response · Authors · 2024-08-12
>
> We appreciate the reviewer for the valuable feedback. We have provided the requested experiments below and hope our responses address the concerns raised. We welcome any further questions.
>
> ---
>
> ### [Q2.] The authors should evaluate on a few more datasets from the spurious correlation domain. ###
>
> As the reviewer requested, we conduct experiments on the MultiNLI dataset, and experiments on the CivilComments dataset are currently running due to time constraints. As shown in the table below, our method effectively works on a text dataset as well. We will include the complete table in the revision.
>
> | MultiNLI | Avg Acc. | Worst-group Acc. |
> |---|---|---|
> | JTT | 80.0 | 70.2 |
> | JTT+Ours | 79.8 | **73.6** |
>
> ---
>
> ### [Q3.]  If compute allows, the authors should compute the ablations (Figure 6) for all other datasets. ###
>
> As the reviewer recommended, we conduct additional ablation studies on the Waterbird, BFFHQ, and CIFAR-10C (1%) datasets. The overall results are consistent with those presented in the main paper.
>
> Specifically, in the additional results, unbiased accuracy across varying $k$ values showed that performance was not sensitive to changes in $k$. Although $k$ = 100  was used in all our experiments, it was not always the optimal choice, as we lacked prior knowledge such as bias severity, and some other $k$ values yielded slightly better performance.
> | SelecMix+Ours | k=50 | k=100 | k=150 | k=200 |
> |---|---|---|---|---|
> | BFFHQ | **68.73 $\pm$ 0.79** | 65.80 $\pm$ 3.12 | 68.53 $\pm$ 1.16 | 67.87 $\pm$ 1.16 |
> | Waterbird | 88.37 $\pm$ 0.70 | 89.67 $\pm$ 0.38 | **89.72 $\pm$ 0.45** | 89.25 $\pm$ 0.32 |
>
> For$\lambda$, using $\lambda$ > 0 has shown to yield robust performance across both low-bias and high-bias datasets; however, setting $lambda$ too high can result in performance degradation. For the $\lambda$, on the highly biased dataset, increasing $\lambda$ results in performance degradation, while on the low-bias dataset, $\lambda=0$ causes a significant performance drop.
> | SelecMix+Ours | lambda=0 | lambda=0.1 | lambda=0.2 | lambda=0.5 |
> |---|---|---|---|---|
> | BFFHQ | **68.67 $\pm$ 1.00** | 65.80 $\pm$ 3.12 | 68.53 $\pm$ 1.23 | 67.47 $\pm$ 1.23 |
> | Waterbird | 78.92 $\pm$ 4.16 | **89.67 $\pm$ 0.38** | 88.01 $\pm$ 0.48 | 85.72 $\pm$ 0.49 |
>
> For epochs, aside from the extremely short training with epoch = 1, we observe that performance is not sensitive to the number of epochs.
> | SelecMix+Ours | epoch=1 | epoch=3 | epoch=5 | epoch=7 | epoch=9 | epoch=11 |
> |---|---|---|---|---|---|---|
> | CIFAR10C-1% | 43.76 $\pm$ 0.67 | 44.79 $\pm$ 0.40 | **46.18 $\pm$ 0.33** | 45.43 $\pm$ 0.61 | 45.20 $\pm$ 0.61 | 45.69 $\pm$ 0.08 |
> | Waterbird | 87.90 $\pm$ 0.32 | 89.56 $\pm$ 0.09 | 89.67 $\pm$ 0.38 | 89.98 $\pm$ 0.21 | **90.03 $\pm$ 0.39** | 89.15 $\pm$ 0.63 |
>
> ---
>
> ### [Q9.]  Why were some baselines omitted from Tables 2 and 3? ###
>
> We conduct evaluations of DCWP on the benchmark datasets presented in Tables 2 and 3. As shown in the table below, our method still outperforms DCWP. (Please note that "Ours (best)" refers to the highest performance achieved in Tables 2 and 3.)
>
> |  | CIFAR10C-20% | CIFAR10C-30% | CIFAR10C-50% | CIFAR10C-70% | CIFAF10C-90% | Waterbird | NICO |
> |---|---|---|---|---|---|---|---|
> | DCWP | 63.37 $\pm$ 1.01 | 67.31 $\pm$ 0.54 | 69.61 $\pm$ 0.21 | 71.54 $\pm$ 0.10 | 71.85 $\pm$ 0.08 | 73.31 $\pm$ 1.78 | 44.98 $\pm$ 1.59 |
> | Ours (best) | **66.67 $\pm$ 0.43** | **68.13 $\pm$ 0.45** | **72.79 $\pm$ 0.38** | **73.56 $\pm$ 0.15** | **73.57 $\pm$ 0.16** | **89.67 $\pm$ 0.38** | **45.69 $\pm$ 1.12** |
>
> ---

---

> > ### Comment · Reviewer_xCjP · 2024-08-13
> >
> > Thank you for the detailed response. I would encourage the authors to characterize (either theoretically, or empirically with synthetic noise) the behavior of their method under mislabeling, which is an important potential failure mode. The authors have addressed most of my other concerns, and I would like to keep my score.

---

> ### Author Response · Authors · 2024-08-13
>
> We are grateful for your insightful feedback and the thoughtful suggestions that have significantly enhanced our work.
>
> ---
>
> ### [Q2.] The authors should evaluate on a few more datasets from the spurious correlation domain. ###
>
> We carried out further experiments on the CivilComments-WILDS dataset. As presented in the table below, the results demonstrate that our method performs well on CivilComments-WILDS. Consequently, our method has shown effective performance across CelebA, MultiNLI, and CivilComments, highlighting its effectiveness on text datasets. We will ensure that the complete table is included in the revision.
>
> | CivilComments | Avg Acc. | Worst-group Acc. |
> |---|---|---|
> | JTT | 92.6 | 63.7 |
> | JTT+Ours | 86.9 | **78.5** |
>
> ---

---

> ### Author Response · Authors · 2024-08-13
>
> ### I would encourage the authors to characterize (either theoretically, or empirically with synthetic noise) the behavior of their method under mislabeling, which is an important potential failure mode ###
>
> We sincerely appreciate the reviewer’s valuable suggestion regarding mislabeled samples, which is indeed an important issue in real-world scenarios.
>
> However, we would like to emphasize that the primary focus of our paper, as well as prior studies on spurious correlations, is to address spurious correlations within training datasets. While considering mislabeled samples in conjunction with spurious correlations is an important future direction, please note that the debiasing community is still facing significant challenges in effectively addressing spurious correlations alone. In fact, addressing both problems together introduces a new and complex challenge in this field.
>
> Therefore, we respectfully ask that if the reviewer’s last concern involves addressing the mislabeling problem simultaneously, we would greatly appreciate reconsideration of this given context.
>
> Once again, we sincerely thank you for your thorough review of our paper and for the valuable feedback provided.
>
> ---

---

### Author Rebuttal · Authors · 2024-08-07

We provide Grad-CAM visualizations [1] for both ERM and ERM+Ours on BFFHQ and Waterbird in Figure 1. We also include example images from the top 100 samples, as ranked by BCSI, in Figure 2.

[1] Selvaraju, Ramprasaath R., et al. "Grad-cam: Visual explanations from deep networks via gradient-based localization." CVPR, 2017.

---

### Decision · Program_Chairs · 2024-09-25

**Decision:**

Accept (poster)

**Comment:**

The manuscript proposes a debiased method to remove the spurious correlation within training dataset. Inspired by the mislabeled sample detection, they proposed a simple and effective method to identify bias-conflicting samples and leverage them to improve the performance. Extensive results on diverse datasets validate the effectiveness of the proposed method.

All the reviewers' concerns have been addressed through rebuttal. Based on this, I recommend to present the paper on the conference.